# Beyond Structure: Invariant Crystal Property Prediction with Pseudo-Particle Ray Diffraction

**Bin Cao**[1,4], **Yang Liu**[2]***Longhan Zhang**[1], **Yifan Wu**[3], **Zhixun Li**[2], **Yuyu Luo**[3],
**Hong Cheng**[2], **Yang Ren**[4]***Tong-Yi Zhang**[1,5]*

[1]Guangzhou Municipal Key Laboratory of Materials Informatics, Advanced Materials Thrust,
The Hong Kong University of Science and Technology (Guangzhou)
[2]The Chinese University of Hong Kong
[3]Data Science and Analytics Thrust, The Hong Kong University of Science and Technology (Guangzhou)
[4]Department of physics, The City University of Hong Kong
[5]Materials Genome Institute, Shanghai University

## Abstract

Crystal property prediction, governed by quantum mechanical principles, is computationally prohibitive to solve exactly for large many-body systems using traditional density functional theory. While machine learning models have emerged as efficient approximations for large-scale applications, their performance is strongly influenced by the choice of atomic representation. Although modern graph-based approaches have progressively incorporated more structural information, they often fail to capture long-range atomic interactions due to finite receptive fields and local encoding schemes. This limitation leads to distinct crystals being mapped to identical representations, hindering accurate property prediction. To address this, we introduce PRDNet that leverages unique reciprocal-space diffraction besides graph representations. To enhance sensitivity to elemental and environmental variations, we employ a data-driven pseudo-particle to generate a synthetic diffraction pattern. PRDNet ensures full invariance to crystallographic symmetries. Extensive experiments are conducted on Materials Project, JARVIS-DFT, and MatBench, demonstrating that the proposed model achieves state-of-the-art performance. The code is openly available at https://github.com/Bin-Cao/PRDNet.

## 1 Introduction

Material properties are fundamentally determined by quantum mechanical equations such as the Schrödinger equation (Kohn et al., 1996), which are intractable for many-body systems and thus commonly approximated by Density Functional Theory (DFT). However, the high computational cost of DFT limits its application to large atomic systems. Recently, machine learning (ML) force fields have emerged as scalable surrogates (Chen et al., 2019; Chen & Ong, 2022), trained on DFT data to enable efficient structure relaxation and property prediction across diverse materials. These models achieve near-DFT accuracy while allowing exploration of large-scale atomic correlations.

In particular, the performance of ML models strongly depends on how atomic systems are represented (Xie & Grossman, 2018; Schütt et al., 2017; Chen et al., 2019; Louis et al., 2020; Choudhary & DeCost, 2021; Yan et al., 2022; 2024; Taniai et al.; Ito et al., 2025). Early efforts relied on hand-engineered descriptors to capture atomic interactions (Isayev et al., 2017; Su et al., 2024; Zhu et al., 2024), which offered interpretability but suffered from human bias and poor transferability. With the emergence of data-driven models, Graph Neural Networks

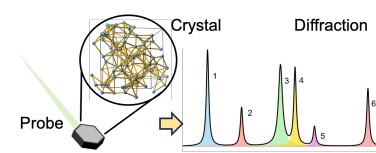

Figure 1: Relationship between crystal and reciprocal space.

*Corresponding authors (yangliu005@cuhk.edu.hk, yangren@cityu.edu.hk, mezhangt@hkust-gz.edu.cn)

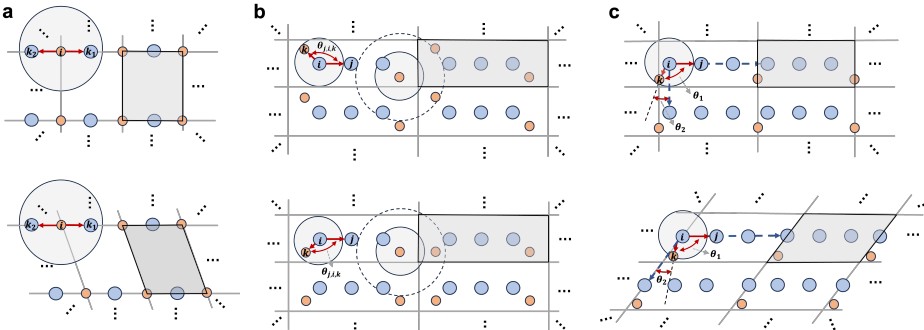

Figure 2: Representation limitations of projecting different atomic periodicity into the same graph representation using (**a**) multi-edge graphs, (**b**) atomic angular embedding, and (**c**) a periodic vector-based reference system. The gray region represents a lattice cell of the 2D crystal.

have become mainstream crystal encoding models. Specifically, they treat crystals as graphs and perform message passing (Chen et al., 2019) to aggregate node information. Initial solutions focus solely on pairwise local interactions, leveraging graph convolution networks (Xie & Grossman, 2018; Chen et al., 2019) and graph attention networks (Louis et al., 2020). To capture higher-order geometry, later works incorporated atomic angular embedding (e.g., bond angles and multi-body descriptors) (Choudhary & DeCost, 2021; Chen & Ong, 2022), improving structural fidelity but at the expense of efficiency and symmetry invariance. Most recently, equivariant/invariant GNNs (Yan et al., 2022; 2024; Taniai et al.; Ito et al., 2025) have emerged as the leading paradigm, systematically overcoming prior limitations by embedding symmetry constraints such as periodicity-preserving message passing, thereby achieving state-of-the-art performance in Crystal Property Prediction (CPP).

However, crystals are, in principle, infinite three-dimensional atomic systems. Current real-space encoders often fail to capture **long-range atomic interactions** (Tantivasadakarn et al., 2024), which are essential for determining physical properties. In particular, these methods typically struggle to represent them, as real-space embeddings with finite receptive fields in multi-edge graph constructions limit their ability to account for long-range effects. As a result, they map distinct crystal structures into the same representation. Figure 2 illustrates these examples. In DFT, these interactions are addressed through boundary conditions and supercell configurations (Makkar & Ghosh, 2021). A promising alternative is diffraction-space encoding: since every atom contributes to diffraction along specific lattice planes, ideal representation is **guaranteed** to embed complete real-space information without loss (Bouwkamp, 2016), enabling more faithful modeling of long-range correlations. Figure 1 illustrates the relationship between the crystal graph and its diffraction. Diffraction arises from the periodic arrangement of atoms and their interactions with the probing beam, described by form factors. Traditional methods, such as X-ray diffraction, rely on fixed, tabulated atomic form factors that depend only on the scattering vector and atomic species, deriving from historically collected, averaged quantum mechanical calculations and experimental measurements. This makes them incapable of distinguishing between atoms of the same element situated in different local chemical environments, as they ignore crucial structural variations that affect material properties.

Reciprocal-space representation is an effective strategy for capturing long-range interactions while remaining compact. This is because the complete diffraction pattern can be analytically derived from a single real-space unit cell, without requiring large supercell constructions, due to the intrinsic periodicity of crystals. However, achieving a lossless and physically valid reciprocal-space representation requires addressing several key gaps that existing works (Kosmala et al., 2023; Lin et al., 2023; Nie et al., 2025) have not systematically resolved. (1) The diffraction result must strictly satisfy the invariance constraints required by crystallographic symmetry. (2) Beyond real physical particles, which are inherently limited by physical laws and therefore not ideal as universal probes, a more flexible and learnable probing mechanism is needed. (3) The form factor must preserve its full physical dependencies, namely atomic species, local chemical environment, and diffraction-vector dependence, which are often simplified or omitted in prior methods. **In addition, diffraction is fundamentally a global property of the entire structure, and therefore must be integrated at the modality level rather than treated as atom-wise feature fusion.**

To exploit this, we introduce a novel pseudo-particle representation, named PRDNet, that serves as a **unique probe**, learned through neural networks, and shown to be more effective for differentiating atoms and structural environments than conventional physical probes such as X-rays, electrons,

or neutrons. Building on this idea, we develop a multimodal framework that integrates both the graph and diffraction information into the learning process. In addition, we show that the proposed model fulfills the geometric constraints, invariant to rotations, reflections, and translations of crystals. Quantitative evaluations on large-scale datasets, including Materials Project, JARVIS-DFT, and MatBench, demonstrate that our method consistently surpasses state-of-the-art models in common crystal property prediction tasks. In summary, our main contributions are as follows :

- We highlight the fundamental challenge of crystal representation, noting that existing methods can map distinct crystal structures to the same representation, and clarify the previously omitted premise of complete reciprocal-space information. To address this, we introduce diffraction-based representations, which are physically guaranteed to be unique.

- We propose PRDNet, a novel architecture that integrates graph embeddings with a learned pseudo-particle diffraction module. It generates synthetic diffraction patterns that are invariant to crystallographic symmetries.

- We extensively evaluate PRDNet on multiple large-scale benchmarks, including Materials Project, JARVIS-DFT, and MatBench. Our model achieves state-of-the-art performance across a wide range of crystal property prediction tasks, demonstrating its effectiveness.

## 2 RELATED WORK

Current crystal representations can be broadly categorized into three types: (1) multi-edge graph models; (2) atomic angular embedding; (3) periodic vector-based reference system. Despite their differences, they share a common limitation: distinct crystal structures may be mapped onto identical representations, as shown in Figure 2. Specifically, multi-edge crystal graphs, as in CGCNN (Xie & Grossman, 2018), SchNet (Schütt et al., 2017), MEGNet (Chen et al., 2019), and GAT-GNN (Louis et al., 2020), define atomic connectivity through cutoff radii, continuous filters, or

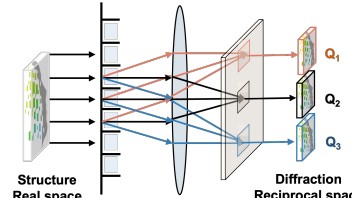

Figure 3: Analogy to light diffraction demonstrating how reciprocal space encodes long-range interactions.

message-passing and attention mechanisms. They are unable to distinguish distinct crystal systems, since angular information is not considered (Figure 2 **a**). To address this, models such as ALIGNN (Choudhary & DeCost, 2021) and M3GNet (Chen & Ong, 2022) incorporate bond-angle features, although restricted cutoffs may still yield identical angular relationships (Figure 2 **b**). Finally, multiple methods extend beyond lattice vectors by introducing periodic vectors, as in MatFormer (Yan et al., 2022) and iConformer/eConformer (Yan et al., 2024), or by employing Transformer-based frameworks, such as Crystalformer (Taniai et al.) and Crystalframer (Ito et al., 2025), to capture broader correlations (Figure 2 **c**). These approaches can satisfy basic crystallographic invariances, but they may still **map distinct crystals to the same local neighborhood**, particularly when the differences are governed by long-range periodicity or subtle symmetry variations.

Recent studies have highlighted the importance of long-range correlations in crystal representations and have sought to capture them through diffraction- or reciprocal-space descriptions. The early attempts aimed to introduce long-range descriptors in reciprocal space to capture the long-range interactions in Coulombic and van der Waals potentials, that are often omitted in ML force fields (Yu et al., 2022). The framework is straightforward: they apply Ewald summation to construct the reciprocal-space potential, which is then concatenated with real-space features to produce more comprehensive predictions. Following this line of work, models such as EwaldMP (Kosmala et al., 2023), PotNet (Lin et al., 2023), and ReGNet (renamed ReciNet in the latest version) (Nie et al., 2025) integrate Ewald summation/Fourier transform into message passing to model long-range interactions. However, existing methods often treat Ewald summation as a Fourier-like information fusion and overlook the unique and invariant nature of form factors, which **should not be propagated** through information aggregation across blocks or layers. Physically, form factors are functional quantities determined solely by the crystal structure and probe; they are unique, determined, and invariant.

In this work, we construct a complete **invariant diffraction representation** based on the fusion of structure and spectrum modalities, rather than atomic interaction information infusion, to explicitly encode long-range interactions. The diffraction-space description reflects the physical process of probing periodic atomic structures with incident particles. As illustrated by the analogy in Figure 3,

the diffraction intensity at a reciprocal vector $\mathbf{Q}$ reflects the aggregated contribution of the entire real-space structure along different diffraction frequencies. That is, the response of an infinitely periodic atomic arrangement along that direction in crystal diffraction. By aggregating neighboring reciprocal vectors in a close group, we obtain multiple directional slices of the structure, which together provide a compact and physically consistent description of long-range interactions. A complete reciprocal-space representation depends on three key factors: (1) periodic atomic types $f_i^{\text{type}}$, (2) diffraction basis $|\mathbf{Q}|$, and (3) atomic local environments $G_\theta(\mathcal{G})$ (representing charge-density distributions), as detailed in Section 4.3. Existing models such as EwaldMP and PotNet approximate infinite-range potentials but ignore $G_\theta(\mathcal{G})$ in EwaldMP and both $G_\theta(\mathcal{G})$ and $|\mathbf{Q}|$ dependencies in PotNet, while ReGNet similarly applies a Fourier-like transformation in each reciprocal block without considering dependencies on $G_\theta(\mathcal{G})$ and $|\mathbf{Q}|$. Such omissions can lead to inconsistent reciprocal representations by violating the structure dependence and invariant of form factors, as discussed in Appendix A.3.

## 3   PRELIMINARIES

**Crystal Structures**   Crystal structures can be described using a fundamental unit, such as a primitive cell, conventional unit cell, Bravais lattice, or asymmetric unit, together with its associated atoms. The unit is defined by a $3 \times 3$ lattice matrix that specifies its geometry in real space. Formally, a crystal structure is represented as $\mathcal{M} = (\mathbf{A}, \mathbf{P}, \mathbf{L})$, where $\mathbf{A}$ denotes atomic types, $\mathbf{P} = [\mathbf{p}_1, \ldots, \mathbf{p}_N] \in \mathbb{R}^{3 \times N}$ contains the fractional or Cartesian coordinates of the $N$ atoms, and $\mathbf{L} = [\boldsymbol{\ell}_1, \boldsymbol{\ell}_2, \boldsymbol{\ell}_3] \in \mathbb{R}^{3 \times 3}$ defines the periodic lattice in real space. A crystal is an infinite periodic structure, expressed as

$$\hat{\mathbf{P}} = \{\hat{\mathbf{p}}_i \mid \hat{\mathbf{p}}_i = \mathbf{p}_i + k_1\boldsymbol{\ell}_1 + k_2\boldsymbol{\ell}_2 + k_3\boldsymbol{\ell}_3,\ k_1, k_2, k_3 \in \mathbb{Z},\ 1 \le i \le N \}, \tag{1}$$

where $\hat{\mathbf{P}}$ denotes the complete set of atomic positions generated by lattice translations. A crystal structure can also be represented as a graph $\mathcal{G} = (\mathcal{V}, \mathcal{E})$, where vertices $\mathcal{V} = \{v_1, v_2, \ldots, v_N\}$ encode the atomic attributes, and edges $\mathcal{E} \subseteq \mathcal{V} \times \mathcal{V}$ represent atomic interactions defined by interatomic distances derived from $\mathbf{P}$ and $\mathbf{L}$, $\hat{\mathbf{v}} = \{\hat{\mathbf{v}}_i \mid \hat{\mathbf{v}}_i = \mathbf{G}(\mathbf{v}_i),\ 1 \le i \le N\}$ where $\hat{\mathbf{v}}$ contains the shared atomic features of atoms occupying symmetry-equivalent sites under the point group operation $\mathbf{G}$.

**Diffraction**   Diffraction refers to the interaction of probe particles with periodic structures (Cao et al.; Guo et al., 2025). In crystals, diffraction maps the real-space atomic arrangement into momentum (reciprocal) space, where each diffraction feature encodes structural information along specific periodic directions. Since diffraction is uniquely determined by the electron distribution for a given probe, and the electron distribution itself is governed by the potential field arising from the infinite atomic arrangement, diffraction provides a **unique representation** of the real-space crystal structure (Kohn et al., 1996; Cao et al., 2025; 2026). This process is governed by the *structure factor* (Greenfield et al., 1971), which describes how atoms scatter incident radiation (e.g., X-rays in XRD). Formally, diffraction is the Fourier transform of the electron density $\rho(\mathbf{r})$ in real space, giving

$$F(\mathbf{Q}) = \int_{\text{unit cell}} \rho(\mathbf{r})\, e^{-i\mathbf{Q} \cdot \mathbf{r}}\, d\mathbf{r}, \tag{2}$$

where $\mathbf{r}$ is the Cartesian position vector and $\mathbf{Q}$ the basis in the reciprocal-space. The electron density $\rho(\mathbf{r})$ reflects the spatial distribution of electrons and governs the dispersion properties of the crystal.

For a crystal of discrete atoms at positions $\mathbf{r}_j$ with atomic form factors $f_j$, the structure factor reduces to (Hubbell et al., 1975)

$$F(\mathbf{Q}) = \sum_{j=1}^{N} f_j(\mathbf{Q})\, e^{-i\mathbf{Q} \cdot \mathbf{r}_j}, \tag{3}$$

where $N$ is the number of atoms in the unit cell. The atomic form factor $f_j(\mathbf{Q})$ specifies the scattering amplitude of atom $j$ and is subject to physical constraints (see Appendix A.3). Thus, each $F(\mathbf{Q})$ encodes the real-space atomic arrangement along a crystallographic direction.

Unlike the Fourier transform recently applied for value extension, latent-space signal processing, or denoising, which decomposes an arbitrary signal into a frequency components to enrich the representation (Jiao et al., 2023; Choromanski et al., 2024; Dong et al., 2024; Gao et al., 2024), the

Fourier transform in crystal diffraction is a **physical scattering process**. It maps the periodic electron density onto a discrete reciprocal lattice and provides a reciprocal view of the crystal structure under a different basis $\mathbf{Q}$, which physically reflects the crystal's directional arrangement rather than its real-space Cartesian coordinates. Although the mathematical foundation is similar, the diffraction transform represents the accumulated phase differences arising from the periodic distribution/arrangement of electrons/atoms. The former yields a general frequency distribution, whereas the latter produces Bragg intensities that are strictly constrained by lattice periodicity and serve as a **physical representation of the underlying crystal structure**.

**Pseudo-particle** Conventional diffraction probes, including X-ray photons, electrons, and neutrons, have intrinsic limitations in reciprocal-space representations (Bouwkamp, 2016). X-rays (Epp, 2016; Cao et al.) and electrons (Glauber & Schomaker, 1953) scatter from the electron cloud, with intensities determined by the atomic form factor $f_j$, making atoms with similar charge nearly indistinguishable. Neutron diffraction (Shull & Wilkinson, 1953), in contrast, relies on nuclear scattering cross-sections, but heavy atoms of similar mass often show weak contrast. In a numerical framework, these limitations can be addressed by introducing a pseudo-particle with learned particle attributes, acting as a unique probe for reciprocal-space crystal representation, enabling differentiation of atomic arrangements in distinct local environments.

**Problem definition** Many material properties can be derived from the fundamental equations of electronic behavior, where the dynamics of electrons are fully determined by the external potential imposed by the atomic arrangement in a crystal (Martin, 2020). Crystal property prediction thus aims to estimate a physical property $y$ from the crystal structure, represented as a tuple $(\mathbf{A}, \mathbf{P}, \mathbf{L})$. The target property $y$ may be continuous ($y \in \mathbb{R}$) in regression tasks or categorical ($y \in \{1, 2, \ldots, C\}$) in classification tasks with $C$ classes.

## 4 METHOD

### 4.1 OVERALL FRAMEWORK

**Motivation for Learned Pseudo-Particles** Unlike conventional X-ray diffraction, which relies on fixed tabulated form factors, our approach introduces a learned pseudo-particle that incorporates environmental dependence, enabling greater sensitivity to chemical environments and elemental differences. In XRD, the interaction between X-ray photons and atoms (i.e., the form factor) is obtained from the International Tables for Crystallography (Prince, 2004). It is a fixed tabulated value that depends on two factors:

$$f_i^{\text{X-ray}} = f_i^{\text{X-ray}}\big(|\mathbf{Q}|, f_i^{\text{type}}\big) \tag{4}$$

where $|\mathbf{Q}|$ is the magnitude of the scattering vector $\mathbf{Q}$ (the reciprocal-space basis), and $f_i^{\text{type}}$ denotes the atomic species. Thus, real physical particles cannot distinguish between identical atoms located in different local environments for a given scattering vector $\mathbf{Q}$. Moreover, the intrinsic physical properties of X-ray photons limit their sensitivity to specific atoms.

This limitation motivates us to design a pseudo-particle that is explicitly sensitive to the local atomic environment while retaining strong discriminability across different elements, making it well-suited for CPP tasks. Unlike real particles, the learned pseudo-particle is not constrained by the physical limitations of X-rays. We define its form factor as:

$$f_i^{\text{Pseudo}} = f_i^{\text{Pseudo}}\big(|\mathbf{Q}|, G_\theta(\mathcal{G}), f_i^{\text{type}}\big), \tag{5}$$

where $G_\theta(\mathcal{G})$ encodes the local chemical environment learned by the graph network.

**Graph embedding** A crystal structure is represented as a graph $\mathcal{G} = (\mathcal{V}, \mathcal{E})$, where vertices $\mathcal{V} = \{v_1, v_2, \ldots, v_N\}$ correspond to atoms one-hot encoded attributes (Xie & Grossman, 2018) and edges $\mathcal{E} \subseteq \mathcal{V} \times \mathcal{V}$ encode atomic interactions within a cutoff radius. Each atom $i$ is characterized by a node feature $\mathbf{h}_i^{(0)} = \text{Embed}(v_i) \in \mathbb{R}^d$ and coordinates $\mathbf{r}_i = (x_i, y_i, z_i) \in \mathbb{R}^3$. Edge features between atoms $i$ and $j$ are defined as:

$$\mathbf{e}_{ij} = \text{RBF}(d_{ij}) \oplus \text{SBF}(\theta_{ijk}) \oplus d_{ij}, \quad d_{ij} = \|\mathbf{r}_i - \mathbf{r}_j\|, \tag{6}$$

where $\text{RBF}(\cdot)$ denotes a radial basis function, $\text{SBF}(\cdot)$ a spherical basis function (Gasteiger et al.), $\theta_{ijk}$ is the angle between bond $ij$ and bond $ik$, and $\oplus$ indicates concatenation.

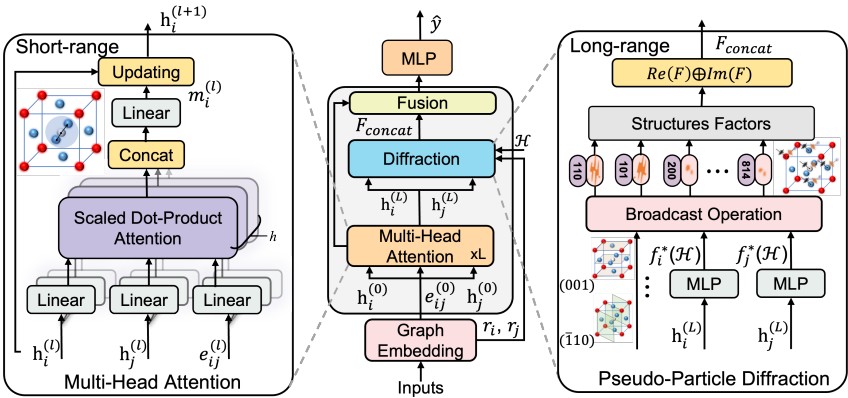

Figure 4: The configuration of PRDNet, which integrates crystal attention and pseudo-particle diffraction to capture both short- and long-range atomic interactions.

**Multi-Modal Feature Fusion** The integration of graph-based crystal real-space features $\mathbf{h}_i^{(L)}$ and physics-based diffraction representations $\mathbf{F}_{\text{concat}}$ is achieved through a fusion mechanism:

$$\mathbf{g} = \text{GlobalPool}(\{\mathbf{h}_i^{(L)}\}_{i=1}^N), \quad \mathbf{d} = \text{MLP}_{\text{diff}}(\mathbf{F}_{\text{concat}}), \quad \mathbf{z}_{\text{fused}} = \text{MLP}_{\text{fusion}}([\mathbf{g} \oplus \mathbf{d}]). \quad (7)$$

GlobalPool$(\cdot)$ aggregates node features across the entire crystal structure, $\text{MLP}_{\text{diff}}(\cdot)$ maps physics-based descriptors to the embedding space, and $\text{MLP}_{\text{fusion}}(\cdot)$ produces the final fused representation.

**Model Invariance** Crystals are inherently symmetrical. Following existing work (Yan et al., 2024; Liu et al., 2024; Zheng et al., 2024), we force the model prediction to be invariant to $E(3)$ group (i.e., the rotation, reflection, and translation of input crystals). Specifically, according to Eq. 7, the final representation output $\mathbf{z}_{\text{fused}}$ depends on the $\mathbf{g}$ and $\mathbf{d}$. Since $\mathbf{g}$ depends on the relative geometric features $d_{ij}$ and $\theta_{ijk}$ (Eq.6) that remain unchanged under $E(3)$ operations, $\mathbf{g}$ is $E(3)$-invariant. Consequently, because $\mathbf{d}$ is also $E(3)$-invariant, the model output is $E(3)$-invariant, as shown in Section 4.3.

## 4.2 STRUCTURE MODELING

**Multi-Head Attention Projections** The PRDNet message-passing layer augments conventional graph attention mechanisms by integrating edge features and physics-informed constraints. Using a multi-head attention framework, it captures information from local atomic environments. Given node features $\mathbf{h}_i^{(l)}, \mathbf{h}_j^{(l)} \in \mathbb{R}^d$ and edge feature $\mathbf{e}_{ij}$ at layer $l$, we first compute the query, key, value, and edge projections for each attention head $h \in \{1, 2, \ldots, H\}$:

$$\mathbf{Q}^{(h)} = \mathbf{W}_Q^{(h)}\mathbf{h}^{(l)}, \ \mathbf{K}^{(h)} = \mathbf{W}_K^{(h)}\mathbf{h}^{(l)}, \ \mathbf{V}^{(h)} = \mathbf{W}_V^{(h)}\mathbf{h}^{(l)}, \ \mathbf{E}_{ij}^{(h)} = \mathbf{W}_E^{(h)}\mathbf{e}_{ij}, \quad (8)$$

where $\mathbf{W}_Q^{(h)}, \mathbf{W}_K^{(h)}, \mathbf{W}_V^{(h)} \in \mathbb{R}^{d_h \times d}$ and $\mathbf{W}_E^{(h)} \in \mathbb{R}^{d_h \times d_e}$ are learnable projection matrices. Then the attention scores are computed as:

$$\mathbf{q}_{ij}^{(h)} = \mathbf{Q}_i^{(h)} \oplus \mathbf{Q}_i^{(h)} \oplus \mathbf{Q}_i^{(h)}, \ \mathbf{k}_{ij}^{(h)} = \mathbf{K}_i^{(h)} \oplus \mathbf{K}_j^{(h)} \oplus \mathbf{E}_{ij}^{(h)}, \ \boldsymbol{\alpha}_{ij}^{(h)} = \frac{\mathbf{q}_{ij}^{(h)} \odot \mathbf{k}_{ij}^{(h)}}{\sqrt{3d_h}}, \quad (9)$$

where $\oplus$ denotes concatenation and $\odot$ represents element-wise multiplication.

**Gated Message and Aggregation** The message aggregation incorporates both value information and attention-modulated gating. For each attention head $h$, the value vector and gating factor are defined as:

$$\mathbf{v}_{ij}^{(h)} = \mathbf{V}_i^{(h)} \oplus \mathbf{V}_j^{(h)} \oplus \mathbf{E}_{ij}^{(h)}, \quad \mathbf{g}_{ij}^{(h)} = \sigma\Big(\text{LayerNorm}(\boldsymbol{\alpha}_{ij}^{(h)})\Big), \quad \mathbf{m}_{ij}^{(h)} = \mathbf{W}_{\text{msg}}\mathbf{v}_{ij}^{(h)} \odot \mathbf{g}_{ij}^{(h)}, \quad (10)$$

where $\sigma(\cdot)$ is the sigmoid activation, $\odot$ denotes elementwise multiplication, and $\mathbf{W}_{\text{msg}} \in \mathbb{R}^{3d_h \times 3d_h}$ is a learnable transformation matrix. The gating term $\mathbf{g}_{ij}^{(h)}$ acts as an adaptive filter, modulating the

contribution of each message according to the attention scores. Messages from all neighbors and attention heads are then aggregated and projected back to the hidden dimension:

$$\mathbf{m}_i^{\text{agg}} = \bigoplus_{j \in \mathcal{N}(i)} \bigoplus_{h=1}^{H} \mathbf{m}_{ij}^{(h)}, \quad \mathbf{h}_i^{(l+1)} = \boldsymbol{\beta}_i \odot \mathbf{h}_i^{(l)} + (1 - \boldsymbol{\beta}_i) \odot \text{SiLU}(\text{BatchNorm}(\mathbf{W}_{\text{concat}} \mathbf{m}_i^{\text{agg}})), \quad (11)$$

where $\mathcal{N}(i)$ denotes the neighbors of node $i$, $\bigoplus$ indicates concatenation, $H$ is the number of attention heads, and $\mathbf{W}_{\text{concat}}$ projects the concatenated features into the hidden dimension.

## 4.3 PSEUDO-PARTICLE RAY DIFFRACTION MODELING

The pseudo-particle ray diffraction module incorporates particle diffraction theory into the neural network architecture. This module computes structure factors from the learned atomic representations and fuses this physical information with graph-based features.

**Learnable Atomic Form Factors** Unlike traditional crystallographic calculations that rely on tabulated atomic form factors, the pseudo-particle form factor $f_i(\mathbf{Q})$ is defined as

$$f_i^*(\mathbf{Q}) = f_i\big(|\mathbf{Q}|, G_\theta(\mathcal{G}), f_i^{\text{type}}\big), \quad (12)$$

where $|\mathbf{Q}|$ depends on the magnitude of the scattering vector $\mathbf{Q}$, $G_\theta(\mathcal{G})$ captures modifications due to the local chemical environment, and $f_i^{\text{type}}$ encodes species-specific attributes. In PRDNet, these adaptive form factors are not fixed but learned through a dedicated form-factor layer that maps node embeddings to scattering strengths:

$$f_i^*(\mathcal{H}) = \text{MLP}_{\text{form}}(\mathbf{h}_i^{(L)}) \in \mathbb{R}^{N_{\text{hkl}}}, \quad (13)$$

where $\text{MLP}_{\text{form}} : \mathbb{R}^d \to \mathbb{R}^{N_{\text{hkl}}}$, and $\mathbf{h}_i^{(L)}$ denotes the final node representation. Let $f_i^*$ denotes the dimension corresponding to a specific $(h, k, l) \in \mathcal{H}$, with $(h, k, l)$ and $\mathcal{H}$ introduced below.

**Miller Index Selection and Structure Factors** To ensure comprehensive coverage of reciprocal space, we employ a systematic selection of Miller indices. Each $(h, k, l)$ triplet specifies a crystallographic direction, corresponding to a unique scattering vector $\mathbf{Q}$.

$$\mathcal{H}_0 = \{(h, k, l) \in \mathbb{Z}^3 : |h|, |k|, |l| \leq C_{\max}, \gcd(|h|, |k|, |l|) = 1\} \quad (14)$$

$\gcd(\cdot)$ denotes the greatest common divisor (GCD). The constraint $\gcd(|h|, |k|, |l|) = 1$ ensures that all fundamental reflections are considered, avoiding redundant higher-order reflections. The parameter $C_{\max} = 8$ controls the resolution in reciprocal space. To guarantee that the index set is closed under symmetry, we define

$$\mathcal{H} = \big\{\pm\text{perm}(h, k, l) : (h, k, l) \in \mathcal{H}_0\big\}, \quad (15)$$

where $\text{perm}(h, k, l)$ denotes all permutations of $(h, k, l)$ and the prefactor $\pm$ accounts for inversion. By construction, $\mathcal{H}$ is closed under all crystallographic operations.

The complete diffraction signature is constructed by computing structure factors for all selected Miller indices $\mathbf{F}(\mathcal{H}) = [F_{hkl}]_{(h,k,l) \in \mathcal{H}}$. The structure factors of pseudo-particle ray diffraction, denoted as $\mathbf{F}(\mathcal{H})$, are converted to real values by applying the real part operator. The real and imaginary components are obtained by summing over atoms with the corresponding form factor weights:

$$\text{Re}(F_{hkl}) = \sum_{i=1}^{N} f_i^* \cos(2\pi \mathbf{h} \cdot \mathbf{r}_i^T), \quad \text{Im}(F_{hkl}) = \sum_{i=1}^{N} f_i^* \sin(2\pi \mathbf{h} \cdot \mathbf{r}_i^T) \quad (16)$$

where $f_i^*$ is the learned atomic form factor for atom $i$, $\mathbf{h} = (h, k, l)^T$ is the reciprocal lattice vector, and $\mathbf{r}_i = (x_i, y_i, z_i)$ represents the fractional coordinates of atom $i$. The real and imaginary parts are then flattened to construct the diffraction feature tensor:

$$\mathbf{F}_{\text{concat}} = \text{flatten}(\text{Re}(\mathbf{F}(\mathcal{H})) \oplus \text{Im}(\mathbf{F}(\mathcal{H}))) \in \mathbb{R}^{2N_{\text{hkl}}} \tag{17}$$

The forward propagation process is presented in Algorithm 1, and a case study on the crystal $CaF_2$ is provided in Appendix C.

**Invariance in pseudo-particle diffraction**   Let $G$ be the crystallographic operations acting on fractional coordinates by $g : \mathbf{r} \mapsto g \cdot \mathbf{r} = R_g \mathbf{r} + \mathbf{t}_g, g \in G$, where $R_g$ (i.e., rotations and reflections) is an integer unimodular matrix ($\det R_g = \pm 1$) that maps Miller indices to integer triplets. And $\mathbf{t}_g$ the translational part. Denote a Miller index by $\mathbf{h} = (h, k, l)^\top \in \mathbb{Z}^3$ and the set of indices by $\mathcal{H}$. **The set $\mathcal{H}$ is closed** under the action of $G$, i.e.,

$$\forall g \in G, \ \forall \mathbf{h} \in \mathcal{H} \quad \Rightarrow \quad g \cdot \mathbf{h} = R_g \mathbf{h} + \mathbf{t}_g = R_g \mathbf{h} \in \mathcal{H}. \tag{18}$$

Under $g$ the coordinates become $g \cdot \mathbf{r}_i = R_g \mathbf{r}_i + \mathbf{t}_g$ and the Miller index maps to $g \cdot \mathbf{h} = R_g \mathbf{h}$. Hence

$$F_{g \cdot \mathbf{h}}(\{g \cdot \mathbf{r}_i\}) = \sum_{i=1}^{N} f_i^* \exp\left(2\pi i \left(R_g \mathbf{h}\right) \cdot \left(R_g \mathbf{r}_i + \mathbf{t}_g\right)\right)$$

$$= \exp\left(2\pi i \left(R_g \mathbf{h}\right) \cdot \mathbf{t}_g\right) \sum_{i=1}^{N} f_i^* e^{2\pi i \mathbf{h} \cdot \mathbf{r}_i} = e^{2\pi i \phi(g, \mathbf{h})} F_{\mathbf{h}}(\{\mathbf{r}_i\}) \tag{19}$$

The phase $\phi(g, \mathbf{h}) = (R_g \mathbf{h}) \cdot \mathbf{t}_g$ arises from the translation. Because it is always an integer, the factor $e^{2\pi i \phi(g,\mathbf{h})} = 1$, which ensures that the representation remains invariant under crystallographic operations.

## 5   EXPERIMENTS

**Datasets**   We benchmark the models on three standard evaluation suites, all independently retrieved from the official websites. The definitions and background of the regression and classification targets are introduced in the Appendix E.

- **Materials Project (MP) Database**: We use stable structures retrieved from the Materials Project (Jain et al., 2020), comprising 122,959 entries with annotated formation energy, band gap, and metal/non-metal classification labels. Additionally, 9,473 of these entries include mechanical properties such as bulk modulus, shear modulus and Young's modulus. The dataset is managed using the *Atomic Simulation Environment (ASE)*, and is available at https://huggingface.co/datasets/caobin/CPPbenchmark.

- **JARVIS-DFT (dft_3d)**: This dataset contains 75,993 entries, each annotated with formation energy, band gap (calculated using either the OptB88vDW functional, denoted as OPT, or the TBMBJ functional, denoted as MBJ), bulk modulus, shear modulus, total energy (calculated using the OptB88vDW functional), and energy above the hull ($E_{\text{hull}}$) (Choudhary et al., 2020). We also evaluate the baselines on the `JARVIS-DFT-3D-2021` dataset (55,723 entries), which serves as an important supplementary benchmark.

- **Matbench (MB) Suite**: We evaluate the models on several tasks from the Matbench suite (Dunn et al., 2020): `matbench_jdft2d` (636 entries; exfoliation energy), `matbench_mp_e_form` (132,752 entries; formation energy), `matbench_log_gvrh` (10,987 entries; shear modulus), and `matbench_dielectric` (4,764 entries; refractive index).

**Implementation details**   The specific hyperparameters used are listed in Appendix D. All models follow the original open-source settings without additional tuning. Model performance is evaluated using the Mean Absolute Error (MAE) and accuracy. All implementations are based on the PyTorch framework (Paszke et al., 2019) and trained on NVIDIA GeForce RTX 3090 GPUs.

Table 1: Mean Absolute Error and Accuracy comparison on the MP.

| Method | Param Num $M$ : million | Form. Energy eV/atom | Bandgap eV | Bulk Modulus log(GPa) | Shear Modulus log(GPa) | Young's Modulus log(GPa) | Metal/Non-metal Acc(%) |
|---|---|---|---|---|---|---|---|
| CGCNN (Xie & Grossman, 2018) | 2.0M | 0.041 | 0.262 | 0.079 | 0.162 | 0.155 | 87.5 |
| SchNet (Schütt et al., 2017) | 0.3M | 0.038 | 0.344 | 0.092 | 0.129 | 0.143 | 82.8 |
| MEGNET (Chen et al., 2019) | 0.2M | 0.053 | 0.307 | 0.134 | 0.206 | 0.189 | 86.6 |
| GATGNN (Louis et al., 2020) | 0.6M | 0.083 | 0.343 | 0.051 | 0.111 | 0.109 | 86.3 |
| Matformer (Yan et al., 2022) | 15.4M | 0.035 | 0.193 | 0.089 | 0.306 | 0.291 | 92.7 |
| Crystalformer (Taniai et al.) | 0.8M | 0.049 | 0.251 | 0.064 | 0.140 | 0.127 | 86.5 |
| Crystalfarmer (Ito et al., 2025) | 0.9M | 0.030 | 0.216 | 0.047 | 0.118 | 0.106 | 87.5 |
| eComFormer (Yan et al., 2024) | 5.6M | 0.033 | 0.153 | 0.088 | 0.269 | 0.311 | 82.5 |
| EwaldMP (Kosmala et al., 2023) | 12.2M | 0.052 | 0.332 | 0.098 | 0.175 | 0.186 | 84.2 |
| PotNet (Lin et al., 2023) | 1.8M | 0.035 | 0.251 | 0.104 | 0.175 | 0.151 | 88.9 |
| ReGNet (Nie et al., 2025)[1] | 3.3M | 0.047 | 0.331 | 0.088 | 0.172 | 0.204 | 85.1 |
| **PRDNet** | 20.9M | **0.028** | **0.151** | **0.035** | **0.108** | **0.104** | **93.3** |

Table 2: Mean Absolute Error comparison on the JARVIS-DFT (left) and MB (right).

| Method | JARVIS-DFT | | | | | | | MB | | | |
|---|---|---|---|---|---|---|---|---|---|---|---|
| | Form. Energy eV/atom | Bandgap (OPT) eV | Bulk Modulus log(GPa) | Shear Modulus log(GPa) | Total Energy eV/atom | Bandgap (MBJ) eV | $E_{hull}$ eV | jdft2d eV/atom | mp_e_form eV/atom | log_gvrh log(GPa) | dielectric unitless |
| CGCNN | 0.038 | 0.144 | 0.125 | 0.163 | 0.041 | 0.402 | 0.049 | 0.050 | 0.027 | 0.066 | 0.541 |
| SchNet | 0.074 | 0.278 | 0.143 | 0.225 | 0.048 | 0.507 | 0.073 | 0.056 | 0.042 | 0.085 | 0.341 |
| MEGNET | 0.074 | 0.221 | 0.124 | 0.214 | 0.054 | 0.369 | 0.056 | 0.052 | 0.036 | 0.098 | 0.459 |
| GATGNN | 0.069 | 0.235 | 0.078 | 0.160 | 0.053 | 0.372 | 0.048 | 0.046 | 0.032 | 0.087 | 0.408 |
| Matformer | 0.033 | 0.207 | 0.101 | 0.157 | 0.041 | 0.341 | **0.034** | 0.072 | 0.023 | 0.071 | 0.764 |
| Crystalformer | 0.052 | 0.248 | 0.075 | 0.146 | 0.055 | 0.282 | 0.053 | 0.071 | 0.020 | 0.080 | 0.244 |
| Crystalframer | 0.048 | 0.208 | 0.065 | 0.144 | 0.047 | 0.270 | 0.050 | 0.056 | 0.065 | 0.084 | 0.248 |
| eComFormer | 0.119 | 0.260 | 0.118 | 0.243 | 0.035 | 0.335 | 0.034 | 0.031 | 0.020 | 0.085 | 0.543 |
| EwaldMP | 0.052 | 0.256 | 0.109 | 0.191 | 0.057 | 0.451 | 0.047 | 0.052 | 0.053 | 0.117 | 0.485 |
| PotNet | 0.034 | 0.144 | 0.112 | 0.156 | 0.039 | 0.339 | 0.079 | 0.033 | 0.029 | 0.084 | 0.397 |
| ReGNet[1] | 0.054 | 0.201 | 0.092 | 0.178 | 0.041 | 0.355 | 0.051 | 0.045 | 0.033 | 0.079 | 0.512 |
| **PRDNet** | **0.032** | **0.140** | **0.064** | **0.122** | **0.032** | **0.267** | 0.041 | 0.038 | **0.019** | **0.058** | **0.242** |

## 5.1 RESULTS

**Benchmarks Performance** Across the MP, JARVIS-DFT, and Matbench benchmarks, PRDNet consistently delivers superior or competitive performance compared with existing models. On the MP dataset (Table 1), PRDNet surpasses all baselines, achieving the lowest errors in formation energy (0.028 eV/atom), band gap (0.151 eV), bulk modulus (0.035 log(GPa)), shear modulus (0.108 log(GPa)), and Young's modulus (0.104 log(GPa)), while also reaching the highest metal/non-metal classification accuracy (93.3%). On JARVIS-DFT (Table 2), PRDNet again outperforms the baselines, achieving the best results on most tasks, including formation energy (0.032 eV/atom), band gap (OPT, 0.140 eV), bulk modulus (0.064 log(GPa)), shear modulus (0.122 log(GPa)), total energy (0.032 eV/atom) and band gap (MBJ, 0.267 eV) while also achieving competitive performance in $E_{hull}$ (0.041 eV). On the Matbench benchmark (Table 2), PRDNet achieves the lowest errors in formation energy (0.019 eV/atom), shear modulus (0.058 log(GPa)) and refractive index (0.242 (unitless)), while maintaining competitive performance in exfoliation energy prediction. PRDNet also achieves SOTA performance on the `JARVIS-DFT-3D-2021` dataset (except for the band gap (OPT) task) outperforming all baseline models (Table 8). These results show that PRDNet generalizes effectively across diverse datasets and provides balanced improvements in both energy-related and mechanical property predictions, underscoring its robustness and versatility for crystal property modeling.

**Ablation Study** To validate the effectiveness of each component in the PRDNet architecture, we perform systematic ablation studies that isolate the contributions of individual modules and design choices. We evaluate PRDNet across diverse property prediction tasks using the Materials Project, JARVIS-DFT and Matbench databases, ensuring a comprehensive assessment. Table 3 presents the ablation results on the MP dataset, while Appendix B, Table 5, and Table 6 report the results on JARVIS-DFT and Matbench, respectively. The results show that each component is essential for optimal performance. Removing the diffraction module causes the obversely performance drop, underscoring its critical role. Likewise, using single-head attention, removing residual connections, or excluding edge features consistently degrades performance, indicating that multi-head attention, residual updates, and edge features collectively strengthen representation learning. These findings confirm the effectiveness of our default architecture, in which all modules work synergistically to achieve the highest accuracy across material property predictions.

---

[1]As of November 12, 2025, the ReGNet (ReciNet) code is unavailable; the results were obtained using our own implementation based on the paper's settings. We use a hidden dimension of 304 instead of 256, as this brings the total parameter count close to the 3.3M reported in the paper; using 256 results in only 2.37M.

Table 3: Ablation study on the Materials Project. **Diff.** indicates whether the diffraction module is used, **MH** denotes single-head or multi-head attention, **Res.** specifies whether residual connections are applied during node attribute updates, and **EF** refers to the inclusion of edge features (see Appendix B).

| Model | Diff. | MH | Res. | EF | Form. Energy MAE(eV/atom) | Bandgap MAE(eV) | Bulk Modulus MAE(log(GPa)) | Shear Modulus MAE(log(GPa)) | Young's Modulus MAE(log(GPa)) | Metal/Non-metal Acc(%) |
|---|---|---|---|---|---|---|---|---|---|---|
| Default | ✓ | ✓ | ✓ | ✓ | **0.028** | **0.151** | **0.035** | **0.108** | **0.104** | **93.3** |
| NoDiff | × | ✓ | ✓ | ✓ | 0.041 | 0.361 | 0.081 | 0.171 | 0.162 | 81.9 |
| SingleHead | ✓ | × | ✓ | ✓ | 0.040 | 0.318 | 0.067 | 0.166 | 0.181 | 89.3 |
| NoRes | ✓ | ✓ | × | ✓ | 0.038 | 0.297 | 0.077 | 0.204 | 0.222 | 82.7 |
| NoEdge | ✓ | ✓ | ✓ | × | 0.043 | 0.355 | 0.071 | 0.198 | 0.179 | 80.1 |

Table 4: Comparison of Mean Absolute Error and Accuracy on the MP dataset across baselines enhanced by diffraction modules. The models were trained on NVIDIA GeForce RTX 4090, GeForce RTX 4080, GeForce RTX 3090, NVIDIA A100 and A40 GPUs.

| Method | GPUs[2] | Form. Energy eV/atom | Bandgap eV | Bulk Modulus log(GPa) | Shear Modulus log(GPa) | Young's Modulus log(GPa) | Metal/Non-metal Acc(%) |
|---|---|---|---|---|---|---|---|
| CGCNN (Xie & Grossman, 2018) | 2×A40 | 0.039 ↓ | 0.207 ↓ | 0.077 ↓ | 0.148 ↓ | 0.150 ↓ | 89.2 ↑ |
| SchNet (Schütt et al., 2017) | 2×A40 | 0.032 ↓ | 0.303 ↓ | 0.091 ↓ | 0.110 ↓ | 0.145(-) | 85.7 ↑ |
| MEGNET (Chen et al., 2019) | 3×3090 | 0.040 ↓ | 0.291 ↓ | 0.087 ↓ | 0.202 ↓ | 0.177 ↓ | 87.1 ↑ |
| GATGNN (Louis et al., 2020) | 2×4080 | 0.062 ↓ | 0.309 ↓ | 0.044 ↓ | 0.169 ↑ | 0.113 (-) | 87.1 ↑ |
| Matformer (Yan et al., 2022) | 2×A100 | 0.034 ↓ | 0.187↓ | 0.082 ↓ | 0.217 ↓ | 0.277 ↓ | 90.1 ↓ |
| Crystalformer (Taniai et al.) | 2×4090 | 0.047 ↓ | 0.252 (-) | 0.059 ↓ | 0.167 ↑ | 0.126 ↓ | 89.6 ↑ |
| Crystalfarmer (Ito et al., 2025) | 3×4080 | 0.032 (-) | 0.166 ↓ | 0.066 ↑ | 0.117 ↓ | 0.108 (-) | 90.7 ↑ |
| eComFormer (Yan et al., 2024) | 2×3090 | 0.032 ↓ | 0.156 (-) | 0.065 ↓ | 0.221 ↓ | 0.287 ↓ | 86.6 ↑ |

**Pseudo-Particle vs. X-ray Photon**  For evidencing the enriched representation of Pseudo-Particle on crystal sensitivity, we compare two PRDNet variants: (1) PRDNet-Learned: employs learned pseudo-particle form factors $f_i^*$ (Eq. 13); (2) PRDNet-Tabulated: substitutes $f_i^*$ with tabulated X-ray photon form factors. As summarized in Table 7 (Appendix B), the pseudo-particle representation consistently outperforms the X-ray counterpart across all datasets and property prediction tasks. Specifically, it achieves lower MAEs for formation energy, bandgap, and mechanical moduli, while also delivering higher classification accuracy in metal/non-metal discrimination.

**Diffraction Module Assembling**  Each real-space crystal representation baseline in Table 1 is integrated with our reciprocal-space diffraction module and the fusion mechanism defined in Eq.,7 and Eqs.,14–17. This allows a systematic comparison of all baseline models after incorporating our diffraction module. Although several baselines are neither invariant nor equivariant, therefore cannot achieve the same level of physical consistency as PRDNet. We include them to provide a more comprehensive evaluation, as listed in Table 4. Overall, the results show that most models benefit from the unique reciprocal-space representation and the modality-level fusion mechanism. Exceptions include GATGNN and Crystalformer on shear modulus, Crystalformer on bulk modulus, and Matformer on metal/non-metal classification, which show decreased performance; all other models show improved or comparable results.

## 6 CONCLUSION

In this work, we introduce a Pseudo-Particle Ray Diffraction Neural Network (PRDNet) for crystal structure property prediction. PRDNet outperforms previous approaches by capturing long-range atomic correlations through a physically invariant reciprocal-space representation embedded within the neural network. The learned pseudo-particle attributes act as a novel probe, exhibiting higher sensitivity than any known physical particle to atomic type, local chemical environment, and diffraction directions. Experiments demonstrate that PRDNet provides a principled framework for integrating crystallographic theory with deep learning, offering a novel conception of invariant crystal representation and achieving SOTA performance across multiple properties and databases. The limitation also arises from the fusion of complete reciprocal-space information, which implies that the inverse process for Crystal Structure Prediction (CSP), i.e., generating valid structures in real space by shaping distributions in diffraction space, can break invariance and thus cannot be directly applied to CSP. In future, we plan to: (1) Integrate PRDNet with the structure identification tool (XQueryer (Cao et al., 2025)) to enable a fully automated materials analysis workflow; (2) Extend PRDNet to simultaneously handle multiple length scales, from atomic to mesoscopic structures; (3) Extend relaxed invariance structures for CSP tasks.

---

[2]As these are supplemental experiments conducted within a limited time window, we were unable to train all models on the same GPUs. Therefore, we explicitly list the GPUs used for training each baseline for this table.

## ACKNOWLEDGMENTS

This work was supported by the Guangzhou–HKUST(GZ) Joint Funding Program (Nos. 2023A03J0003 and 2023A03J0103). Y. R. also acknowledges financial support from the City University of Hong Kong (Project Nos. 9610533 and 9680351). Y. L. is supported in part by the Postdoctoral Fellowship Scheme of The Chinese University of Hong Kong.

## REPRODUCIBILITY STATEMENT

We are committed to ensuring the reproducibility of our research. To this end, we provide a publicly accessible GitHub repository accompanied by a comprehensive README file. The README provides detailed information on the PRDNet model, its hyperparameters, environment configuration, dependency installation, environment validation, datasets, training executor, and training procedures. Furthermore, Appendix D lists all benchmark model settings used for validation.

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

# A    ADDITIONAL DETAILS

## A.1    THE USE OF LLMS

In this work, we used LLMs as an auxiliary tool for polishing the writing and improving the clarity of the manuscript. Specifically, we use it to review the entire text for grammatical accuracy, improve sentence structure, and ensure consistent phrasing and tone throughout the paper.

## A.2    CRYSTAL INVARIANCE

The representation of a crystal structure must satisfy several fundamental **invariance properties** to reflect the physical symmetries inherent in periodic materials (Li et al., 2025). These include:

- **Permutation invariance**: Let $f(\mathcal{G})$ denote a representation of a crystal graph $\mathcal{G} = (\mathcal{V}, \mathcal{E})$, where $\mathcal{V}$ is the set of atoms and $\mathcal{E}$ the set of edges. Then, for any permutation $\pi$ of the node indices, we require

$$f(\mathcal{G}) = f(\pi(\mathcal{G})), \tag{20}$$

  meaning that relabeling atoms does not alter the representation. This property is naturally satisfied in graph-based models.

- **Translation invariance**: If all atomic positions $\mathbf{r}$ are shifted by a vector $\mathbf{t} \in \mathbb{R}^3$, i.e., $\mathbf{r} \to \mathbf{r} + \mathbf{t}$, then

$$f(\{\mathbf{r}\}) = f(\{\mathbf{r} + \mathbf{t}\}), \tag{21}$$

  ensuring the representation is invariant under global translations.

- **Rotation invariance**: For any rotation matrix $R \in \mathrm{SO}(3)$, the representation must satisfy

$$f(\{\mathbf{r}_i\}) = f(\{R\mathbf{r}_i\}), \tag{22}$$

  i.e., the representation remains unchanged under rigid body rotations.

- **Periodic invariance**: For any lattice translation vector $\mathbf{T} \in \mathbb{Z}^3$, the atomic positions $\mathbf{r}_i$ and $\mathbf{r}_i + \mathbf{T}$ should yield the same representation:

$$f(\{\mathbf{r}_i\}) = f(\{\mathbf{r}_i + \mathbf{T}\}), \tag{23}$$

  capturing the periodicity of the crystal lattice.

These four invariance properties are essential for any representation of matter based on a periodic unit cell. Beyond these, **symmetry invariance** plays a crucial role in crystallographic systems.

**Symmetry invariance** requires that atoms related by a space group symmetry operation, such as those occupying the same Wyckoff position, must be treated equivalently in the representation. If a symmetry operation $G$ maps atomic coordinates $\mathbf{r}$ to $G(\mathbf{r})$, then the representation must satisfy

$$f(\mathbf{r}) = f(G(\mathbf{r})). \tag{24}$$

Such symmetry operations include:

- **Identity** ($E$): $G(\mathbf{r}) = \mathbf{r}$. This operation is always present by group theory.
- **Inversion** ($\bar{1}$): $G(\mathbf{r}) = -\mathbf{r}$. Crystals exhibiting this symmetry are called *centrosymmetric*.
- **Rotation** ($C_n$): $G(\mathbf{r}) = R_n\mathbf{r}$, where $R_n$ is a rotation matrix corresponding to a $360°/n$ rotation about a symmetry axis.
- **Mirror reflection** ($\sigma$): $G(\mathbf{r}) = \mathbf{r} - 2(\mathbf{r} \cdot \hat{n})\hat{n}$, reflecting the position across a mirror plane with normal vector $\hat{n}$.

In crystallography, certain symmetry operations combine rotation or reflection with fractional lattice translations, giving rise to **glide planes** and **screw axes** (Fjellvåg, 1994). These compound operations are essential for defining space groups but are not directly relevant to the basis invariance required for representation learning.

### A.3 Physical Constraints of Diffraction

Recent works endeavor to introduce Fourier transforms to capture long-range interactions (Kosmala et al., 2023; Lin et al., 2023; Nie et al., 2025), incorporating reciprocal-space information to model long-range atomic correlations to some extent. However, a complete diffraction representation is uniquely invariant and physically determined; slight omissions or approximations may lead to incomplete information, often treated empirically as a form of supplemental atomic information fusion. In our work, we systematically address these dependencies and construct an framework that preserves invariance. Taking ReGNet (renamed ReciNet in the latest version) (Nie et al., 2025) as an example, the model introduces a dedicated reciprocal block that approximates the structure factor, functioning similarly to Eq. 3, and achieves improved performance. We further decompose their key Fourier-transform settings to discuss the underlying dependencies.

- **Dependence:** The atomic form factor $f_j(\mathbf{Q})$, as defined in Eq. 3, is inherently depends on three fundamental factors: the atomic species, the local atomic environment, and the scattering vector $\mathbf{Q}$. Preserving these dependencies is crucial for maintaining physical fidelity in diffraction-based modeling.
- **Consistency:** According to the Schrodinger equation(Berezin & Shubin, 2012), the charge density (or electron distribution) is uniquely determined by the atomic configuration. Therefore, the reciprocal-space representation must remain consistent with this determinism and **cannot** evolve independently.

In ReGNet, the reciprocal embedding is computed as:

$$F(\mathbf{Q}) = \sum_j h_j^\ell \cdot e^{-i\mathbf{Q}\cdot\mathbf{r}_j} \tag{25}$$

where $h_j^\ell$ is a learnable embedding that evolves through network layers. The following dependencies and invariances may be omitted in such formulations:

- **Form factor dependence:** The complete diffraction factor $f_j(\mathbf{Q})$ should be expressed as

$$f_j(\mathbf{Q}) = f_j\big(|\mathbf{Q}|, G_\theta(\mathcal{G}), f_j^{\text{type}}\big), \tag{26}$$

  where $|\mathbf{Q}|$ captures the dependence on the magnitude of the scattering vector, $G_\theta(\mathcal{G})$ reflects the local atomic environment, and $f_j^{\text{type}}$ encodes the atomic species. The $h_j^\ell$ in Eq. 25 loses the dependence on $\mathbf{Q}$ and the local state $G_\theta(\mathcal{G})$, depending only on the atom and block orders.
- **Reciprocal-space uniqueness:** The Fourier transform $F(\mathbf{Q})$ of the charge density $\rho(\mathbf{r})$ is uniquely defined by a given atomic configuration. Eq. 25 evolves the diffraction expression across blocks in a way that breaks this invariance, thereby losing the physical uniqueness.
- **Modal fusion:** Diffraction captures long-range interactions by introducing complete reciprocal-space information, which reflects the periodic arrangement and physical interactions between the structure (i.e., electron attributes) and probing process. This reciprocal-space information is a holistic representation of the structure and thus **should be fused at the modality level rather than at the atomic level.**

### A.4 Training Settings of PRDNET

#### A.4.1 Huber Loss for Outlier Robustness

To accommodate the diverse nature of crystal property prediction tasks and improve training stability, PRDNet employs Huber loss (Meyer, 2021) for training.

The Huber loss combines the advantages of L2 loss for small errors and L1 loss for large errors:

$$\mathcal{L}_{\text{Huber}}(\hat{y}, y; \delta) = \begin{cases} \frac{1}{2}(\hat{y}-y)^2 & \text{if } |\hat{y}-y| \leq \delta \\ \delta|\hat{y}-y| - \frac{1}{2}\delta^2 & \text{otherwise} \end{cases} \tag{27}$$

where $\delta > 0$ is a hyperparameter that controls the transition point between quadratic and linear regimes. This formulation provides robustness against outliers while maintaining smooth gradients for small errors.

### A.4.2 ADAMW WITH DECOUPLED WEIGHT DECAY

PRDNet employs the AdamW optimizer, which decouples weight decay from gradient-based updates:

$$\mathbf{m}_t = \beta_1 \mathbf{m}_{t-1} + (1 - \beta_1)\nabla_{\boldsymbol{\theta}}\mathcal{L}_t \tag{28}$$

$$\mathbf{v}_t = \beta_2 \mathbf{v}_{t-1} + (1 - \beta_2)(\nabla_{\boldsymbol{\theta}}\mathcal{L}_t)^2 \tag{29}$$

$$\hat{\mathbf{m}}_t = \frac{\mathbf{m}_t}{1 - \beta_1^t}, \quad \hat{\mathbf{v}}_t = \frac{\mathbf{v}_t}{1 - \beta_2^t} \tag{30}$$

$$\boldsymbol{\theta}_{t+1} = \boldsymbol{\theta}_t - \alpha_t \left( \frac{\hat{\mathbf{m}}_t}{\sqrt{\hat{\mathbf{v}}_t} + \epsilon} + \lambda\boldsymbol{\theta}_t \right) \tag{31}$$

where $\beta_1 = 0.9$, $\beta_2 = 0.999$, $\epsilon = 10^{-8}$, and $\lambda$ is the weight decay coefficient.

### A.4.3 COSINE ANNEALING WITH WARM RESTARTS

The learning rate schedule follows a cosine annealing pattern with periodic restarts:

$$\eta_t = \eta_{\min} + \frac{1}{2}(\eta_{\max} - \eta_{\min}) \left( 1 + \cos\left( \frac{T_{\text{cur}}}{T_{\max}}\pi \right) \right) \tag{32}$$

$$T_{\text{cur}} = t \bmod T_{\max} \tag{33}$$

where $T_{\text{cur}}$ is the current epoch within the restart cycle, $T_{\max}$ is the restart period, $\eta_{\max}$ is the maximum learning rate, and $\eta_{\min}$ is the minimum learning rate.

### A.4.4 LAYER NORMALIZATION WITH LEARNABLE PARAMETERS

Layer normalization is applied to stabilize training:

$$\mu = \frac{1}{d}\sum_{i=1}^{d} x_i \tag{34}$$

$$\sigma^2 = \frac{1}{d}\sum_{i=1}^{d} (x_i - \mu)^2 \tag{35}$$

$$\text{LayerNorm}(\mathbf{x}) = \frac{\mathbf{x} - \mu}{\sqrt{\sigma^2 + \epsilon}} \odot \boldsymbol{\gamma} + \boldsymbol{\beta} \tag{36}$$

where $\boldsymbol{\gamma}$ and $\boldsymbol{\beta}$ are learnable scale and shift parameters.

## B ABLATION STUDY

### B.1 DIFFRACTION ABLATION

**PRDNET-NoDiff** removes diffraction integration entirely:

$$\mathbf{z}_{\text{nodiff}} = \text{MLP}_{\text{out}}(\mathbf{g}) \tag{37}$$

**PRDNET-Full** incorporates complete diffraction physics integration:

$$\mathbf{z}_{\text{fused}} = \text{MLP}_{\text{fusion}}([\mathbf{g} \oplus \mathbf{d}]) \tag{38}$$

$$\text{where } \mathbf{d} = \text{MLP}_{\text{diff}}([\mathbf{F}_{\text{real}} \oplus \mathbf{F}_{\text{imag}}]) \tag{39}$$

### B.2 ATTENTION MECHANISM ABLATION

**PRDNet-SingleHead** uses single-head attention:

$$\mathbf{m}_{ij}^{\text{single}} = \text{Attention}(\mathbf{Q}_i, \mathbf{K}_j, \mathbf{V}_j, \mathbf{E}_{ij}) \tag{40}$$

**PRDNet-Full**   employs multi-head attention:

$$\mathbf{m}_{ij}^{\text{multi}} = \bigoplus_{h=1}^{H} \text{Attention}^{(h)}(\mathbf{Q}_i^{(h)}, \mathbf{K}_j^{(h)}, \mathbf{V}_j^{(h)}, \mathbf{E}_{ij}^{(h)}) \tag{41}$$

### B.3   RESIDUAL CONNECTION ABLATION

**PRDNet-NoRes**   removes residual connections:

$$\mathbf{h}_i^{(l+1)} = \text{SiLU}(\text{BatchNorm}(\tilde{\mathbf{h}}_i^{(l+1)})) \tag{42}$$

**PRDNet-Full**   uses residuals:

$$\mathbf{h}_i^{(l+1)} = \boldsymbol{\beta}_i \odot \mathbf{h}_i^{(l)} + (1 - \boldsymbol{\beta}_i) \odot \text{SiLU}(\text{BatchNorm}(\tilde{\mathbf{h}}_i^{(l+1)})) \tag{43}$$

### B.4   EDGE FEATURE ABLATION

**PRDNet-NoEdge**   excludes edge features from attention:

$$\boldsymbol{\alpha}_{ij} = \frac{\mathbf{Q}_i \odot \mathbf{K}_j}{\sqrt{d}} \tag{44}$$

**PRDNet-Full**   incorporates edge features:

$$\boldsymbol{\alpha}_{ij} = \frac{(\mathbf{Q}_i \oplus \mathbf{Q}_i \oplus \mathbf{Q}_i) \odot (\mathbf{K}_i \oplus \mathbf{K}_j \oplus \mathbf{E}_{ij})}{\sqrt{3d}} \tag{45}$$

Table 5: Ablation study on the JARVIS-DFT. **Diff.** indicates whether the diffraction module is used, **MH** denotes single-head or multi-head attention, **Res.** specifies whether residual connections are applied during node attribute updates, and **EF** refers to the inclusion of edge features

| Model | Diff. | MH | Res. | EF | Form. Energy | Bandgap | Bulk Modulus | Shear Modulus |
|---|---|---|---|---|---|---|---|---|
| Default | ✓ | ✓ | ✓ | ✓ | **0.032** | **0.140** | **0.064** | **0.122** |
| NoDiff | × | ✓ | ✓ | ✓ | 0.055 | 0.333 | 0.287 | 0.291 |
| SingleHead | ✓ | × | ✓ | ✓ | 0.071 | 0.291 | 0.236 | 0.311 |
| NoRes | ✓ | ✓ | × | ✓ | 0.048 | 0.397 | 0.201 | 0.237 |
| NoEdge | ✓ | ✓ | ✓ | × | 0.083 | 0.412 | 0.289 | 0.341 |

Table 6: Ablation study on the MB. **Diff.** indicates whether the diffraction module is used, **MH** denotes single-head or multi-head attention, **Res.** specifies whether residual connections are applied during node attribute updates, and **EF** refers to the inclusion of edge features

| Model | Diff. | MH | Res. | EF | Exfo. Energy | Form. Energy | Shear Modulus | Refractive Index |
|---|---|---|---|---|---|---|---|---|
| Default | ✓ | ✓ | ✓ | ✓ | **0.038** | **0.019** | **0.058** | **0.242** |
| NoDiff | × | ✓ | ✓ | ✓ | 0.072 | 0.042 | 0.121 | 0.297 |
| SingleHead | ✓ | × | ✓ | ✓ | 0.121 | 0.038 | 0.094 | 0.298 |
| NoRes | ✓ | ✓ | × | ✓ | 0.064 | 0.038 | 0.101 | 0.311 |
| NoEdge | ✓ | ✓ | ✓ | × | 0.099 | 0.051 | 0.099 | 0.304 |

**Number of Miller Indices :**   Each Miller index specifies a crystallographic direction. The choice of a sufficiently large set of Miller indices, $\mathcal{H}_0$, is crucial for adequately representing reciprocal space. We evaluate the performance of PRDNet using 100, 200, 300, 500, and 1000 Miller indices on the MP dataset for both formation energy and band gap prediction tasks. As shown in Figure 5, using 300 indices provides a good balance between computational cost and predictive accuracy.

Table 7: Pseudo-Particle v.s. X-ray Photon

| Method | MP | | JARVIS-DFT | | MB | |
|---|---|---|---|---|---|---|
| | Pseudo-Particle | X-ray | Pseudo-Particle | X-ray | Pseudo-Particle | X-ray |
| **Form. Energy** (MAE eV/atom) | 0.028 | 0.085 | 0.032 | 0.103 | 0.019 | 0.063 |
| **Bandgap** (MAE eV) | 0.151 | 0.271 | 0.140 | 0.331 | - | - |
| **Bulk Modulus** (MAE log(GPa)) | 0.035 | 0.094 | 0.064 | 0.101 | - | - |
| **Shear Modulus** (MAE log(GPa)) | 0.108 | 0.322 | 0.122 | 0.256 | 0.058 | 0.111 |
| **Young's Modulus** (MAE log(GPa)) | 0.104 | 0.291 | - | - | - | - |
| **Exfo. Energy** (MAE eV/atom) | - | - | - | - | 0.038 | 0.089 |
| **Refractive Index** (MAE unitless) | - | - | - | - | 0.242 | 0.476 |
| **Metal/Non-metal** (Acc %) | 93.3 | 78.9 | - | - | - | - |

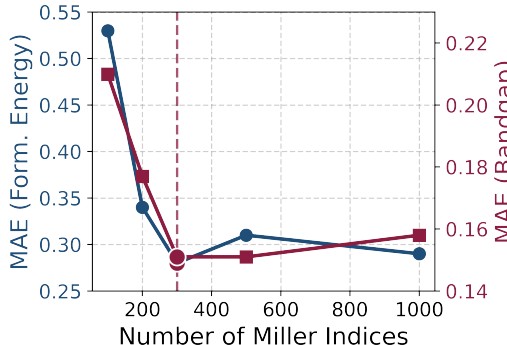

Figure 5: Performance of PRDNet on formation energy and band gap prediction across different numbers of Miller indices.

## C  CASE STUDY OF FORWARD PROPAGATION

To illustrate the complete PRDNet forward propagation process, we present a detailed case using a concrete example crystal structure. This section demonstrates how atomic-level information flows through the network to produce property predictions.

**Input Crystal Structure**  Consider a simple cubic crystal structure $\mathcal{X}$ :

- **Composition**: $CaF_2$

- **Unit cell**: 3 atoms with fractional coordinates

- **Lattice parameters**: $a = b = c = 5.46$ Å, $\alpha = \beta = \gamma = 90°$

- **Space group**: Fm$\bar{3}$m (face-centered cubic)

$$\text{Atom 1 (A):} \quad Z_1 = 20, \quad \mathbf{r}_1 = (0.0, 0.0, 0.0) \tag{46}$$
$$\text{Atom 2 (B):} \quad Z_2 = 9, \quad \mathbf{r}_2 = (0.25, 0.25, 0.25) \tag{47}$$
$$\text{Atom 3 (C):} \quad Z_3 = 9, \quad \mathbf{r}_3 = (0.75, 0.75, 0.75) \tag{48}$$

where $Z_i$ represents the atomic number and $\mathbf{r}_i$ are fractional coordinates.

### C.1  STEP 1: GRAPH CONSTRUCTION AND FEATURE INITIALIZATION

**Neighbor Detection**  Using a cutoff radius $r_{\text{cut}} = 5.0$ Å(for illustration), we identify atomic neighbors:

$$\mathcal{N}(1) = \{2, 3\} \quad \text{(A atom has 1 B + 1 C neighbor)} \tag{49}$$
$$\mathcal{N}(2) = \{1, 3\} \quad \text{(B atom has 1 A + 1 C neighbor)} \tag{50}$$
$$\mathcal{N}(3) = \{1, 2\} \quad \text{(C atom has 1 A + 1 B neighbor)} \tag{51}$$

**Node Feature Initialization** Atomic embeddings are initialized based on atomic numbers:

$$\mathbf{h}_1^{(0)} = \text{Embed}(Z_1 = 20) \in \mathbb{R}^{256} \quad \text{(Ca embedding)} \tag{52}$$

$$\mathbf{h}_2^{(0)} = \text{Embed}(Z_2 = 9) \in \mathbb{R}^{256} \quad \text{(F embedding)} \tag{53}$$

$$\mathbf{h}_3^{(0)} = \text{Embed}(Z_3 = 9) \in \mathbb{R}^{256} \quad \text{(F embedding)} \tag{54}$$

**Edge Feature Construction** For each atomic pair $(i, j)$, edge features encode geometric relationships:

$$d_{12} = \|\mathbf{r}_1 - \mathbf{r}_2\| = 2.36 \text{ Å} \tag{55}$$

$$\mathbf{e}_{12} = \text{RBF}(d_{12}) \oplus \text{SBF}(\theta_{123}) \oplus d_{12} \in \mathbb{R}^{128} \tag{56}$$

where $\text{RBF}(\cdot)$ is the radial basis function and $\text{SBF}(\cdot)$ is the spherical basis function (Gasteiger et al.),.

## C.2 STEP 2: MULTI-LAYER GRAPH CONVOLUTION

For each layer $l = 0, 1, \ldots, L - 1$ (with $L = 6$ layers):

**Multi-Head Attention** For attention head $h = 1$ and atomic pair $(1, 2)$:

$$\mathbf{Q}_1^{(1)} = \mathbf{W}_Q^{(1)} \mathbf{h}_1^{(l)} \in \mathbb{R}^{32} \quad \text{(Query for atom 1)} \tag{57}$$

$$\mathbf{K}_1^{(1)} = \mathbf{W}_K^{(1)} \mathbf{h}_1^{(l)} \in \mathbb{R}^{32} \quad \text{(Key for atom 1)} \tag{58}$$

$$\mathbf{K}_2^{(1)} = \mathbf{W}_K^{(1)} \mathbf{h}_2^{(l)} \in \mathbb{R}^{32} \quad \text{(Key for atom 2)} \tag{59}$$

$$\mathbf{V}_1^{(1)} = \mathbf{W}_V^{(1)} \mathbf{h}_1^{(l)} \in \mathbb{R}^{32} \quad \text{(Value for atom 1)} \tag{60}$$

$$\mathbf{V}_2^{(1)} = \mathbf{W}_V^{(1)} \mathbf{h}_2^{(l)} \in \mathbb{R}^{32} \quad \text{(Value for atom 2)} \tag{61}$$

$$\mathbf{E}_{12}^{(1)} = \mathbf{W}_E^{(1)} \mathbf{e}_{12} \in \mathbb{R}^{32} \quad \text{(Edge projection)} \tag{62}$$

**Attention Score Calculation**

$$\mathbf{q}_{12}^{(1)} = \mathbf{Q}_1^{(1)} \oplus \mathbf{Q}_1^{(1)} \oplus \mathbf{Q}_1^{(1)} \in \mathbb{R}^{96} \tag{63}$$

$$\mathbf{k}_{12}^{(1)} = \mathbf{K}_1^{(1)} \oplus \mathbf{K}_2^{(1)} \oplus \mathbf{E}_{12}^{(1)} \in \mathbb{R}^{96} \tag{64}$$

$$\boldsymbol{\alpha}_{12}^{(1)} = \frac{\mathbf{q}_{12}^{(1)} \odot \mathbf{k}_{12}^{(1)}}{\sqrt{96}} \in \mathbb{R}^{96} \tag{65}$$

**Message Aggregation**

$$\mathbf{m}_{12}^{(1)} = \mathbf{W}_{\text{msg}}(\mathbf{V}_1^{(1)} \oplus \mathbf{V}_2^{(1)} \oplus \mathbf{E}_{12}^{(1)}) \odot \sigma(\boldsymbol{\alpha}_{12}^{(1)}) \tag{66}$$

$$\mathbf{m}_1^{\text{agg}} = \mathbf{m}_{12}^{(1)} \oplus \mathbf{m}_{13}^{(1)} \oplus \cdots \oplus \mathbf{m}_{12}^{(H)} \oplus \mathbf{m}_{13}^{(H)} \tag{67}$$

**Node Updating**

$$\mathbf{h}_1^{(l+1)} = \boldsymbol{\beta}_i \odot \mathbf{h}_i^{(l)} + (1 - \boldsymbol{\beta}_i) \odot \text{SiLU}(\text{BatchNorm}(\mathbf{W}_{\text{concat}} \mathbf{m}_1^{\text{agg}})) \tag{68}$$

## C.3 STEP 3: PHYSICS INTEGRATION - DIFFRACTION MODULE

**Learnable Form Factor** After $L$ graph convolution layers, compute atomic form factors:

$$f_1(\mathcal{H}) = \text{MLP}_{\text{form}}(\mathbf{h}_1^{(L)}) = (2.34, \ldots) \quad \text{(Ca form factor)} \tag{69}$$

$$f_2(\mathcal{H}) = \text{MLP}_{\text{form}}(\mathbf{h}_2^{(L)}) = (1.87, \ldots) \quad \text{(F form factor)} \tag{70}$$

$$f_3(\mathcal{H}) = \text{MLP}_{\text{form}}(\mathbf{h}_3^{(L)}) = (1.91, \ldots) \quad \text{(F form factor)} \tag{71}$$

**Structure Factor** For selected Miller indices, e.g., $(h, k, l) = (1, 0, 0) \in \mathcal{H}$:

$$F_{100} = \sum_{j=1}^{3} f_j \exp(2\pi i(1 \cdot x_j + 0 \cdot y_j + 0 \cdot z_j)) \tag{72}$$

$$= f_1 \exp(2\pi i \cdot 0) + f_2 \exp(2\pi i \cdot 0.25) + f_3 \exp(2\pi i \cdot 0.75) \tag{73}$$

$$= 2.34 + 1.87 e^{i\pi/2} + 1.91 e^{i3\pi/2} \tag{74}$$

$$= 2.34 + 1.87i - 1.91i = 2.34 + (-0.04)i \tag{75}$$

where $i$ is the imaginary unit.

**Diffraction Feature Vector** For $N_{\text{hkl}} = 300$ Miller indices:

$$\mathbf{F}_{\text{real}} = [\text{Re}(F_{100}), \text{Re}(F_{010}), \ldots] = [2.34, 1.23, \ldots] \tag{76}$$

$$\mathbf{F}_{\text{imag}} = [\text{Im}(F_{100}), \text{Im}(F_{010}), \ldots] = [-0.04, 0.87, \ldots] \tag{77}$$

$$\mathbf{F}_{\text{concat}} = [\mathbf{F}_{\text{real}} \oplus \mathbf{F}_{\text{imag}}] \tag{78}$$

### C.4 STEP 4: MULTI-MODAL FUSION

**Graph Feature Pooling**

$$\mathbf{g} = \frac{1}{3}(\mathbf{h}_1^{(L)} + \mathbf{h}_2^{(L)} + \mathbf{h}_3^{(L)}) \in \mathbb{R}^{256} \quad \text{(Mean pooling)} \tag{79}$$

**Diffraction Processing**

$$\mathbf{d} = \text{MLP}_{\text{diff}}(\mathbf{F}_{\text{concat}}) \in \mathbb{R}^{256} \quad \text{(Diffraction features)} \tag{80}$$

**Feature Fusion and Prediction**

$$\mathbf{z}_{\text{fused}} = \text{MLP}_{\text{fusion}}([\mathbf{g} \oplus \mathbf{d}]) \in \mathbb{R}^{128} \tag{81}$$

$$\hat{y} = \text{MLP}_{\text{out}}(\mathbf{z}_{\text{fused}}) = -2.47 \text{ eV/atom} \tag{82}$$

The final prediction $\hat{y} = -2.47$ eV/atom represents the formation energy of the $CaF_2$ crystal structure.

### C.5 EXPERIMENTS ON JARVIS-DFT-3D-2021

In Table 8, we present the performance of PRDNet and the baseline models on the `JARVIS-DFT-3D-2021` dataset. All baseline models were independently retrained for a fair comparison. PRDNet consistently outperforms the baselines, achieving the lowest MAE in formation energy (0.025 eV/atom), bulk modulus (0.075 log(GPa)), shear modulus (0.132 log(GPa)), total energy (0.024 eV/atom), band gap (MBJ, 0.242 eV), and $E_{\text{hull}}$ (0.040 eV). It also demonstrates competitive performance on the band gap (OPT) prediction task, with a MAE of 0.138 eV.

Furthermore, Table 9 summarizes the reused baseline MAE results reported in the respective original papers. PRDNet achieves the best performance on formation energy, total energy, and $E_{\text{hull}}$, while ranking second on band gap (MBJ) and delivering competitive results on band gap (OPT).

### C.6 EXPERIMENTS ON MATERIALS PROJECT (MEGNET) DATASET

In Table 10, we summarize the performance of PRDNet and the baseline models on the Materials Project (MEGNet) dataset. PRDNet achieves the lowest MAE for bulk modulus (0.031 log(GPa)) and shear modulus (0.062 log(GPa)), and a competitive result for band gap (0.187 eV).

---

**Algorithm 1** PRDNet Forward Propagation

---

**Require:** Crystal structure graph $\mathcal{G} = (\mathcal{V}, \mathcal{E})$, atomic positions $\{\mathbf{r}_i\}_{i=1}^{N}$, lattice parameters
**Ensure:** Property prediction $\hat{y} \in \mathbb{R}$

  **Initialize:** Node features $\mathbf{h}_i^{(0)} = \text{Embed}(Z_i)$ for $i = 1, \ldots, N$
  **Compute:** Edge features $\mathbf{e}_{ij} = \text{RBF}(d_{ij}) \oplus \text{SBF}(\theta_{ijk}) \oplus d_{ij}$
  **for** $l = 0$ to $L - 1$ **do**
    **for** each attention head $h = 1$ to $H$ **do**
      Compute projections: $\mathbf{Q}_i^{(h)}, \mathbf{K}_j^{(h)}, \mathbf{V}_j^{(h)}, \mathbf{E}_{ij}^{(h)}$ *(Eq. 8)*
      Calculate attention: $\boldsymbol{\alpha}_{ij}^{(h)}$ *(Eq. 9)*
      Compute messages: $\mathbf{m}_{ij}^{(h)}$ *(Eq. 10)*
    **end for**
    Aggregate multi-head messages: $\mathbf{m}_i^{\text{agg}}$ *(Eq. 11)*
    Apply linear projection: $\tilde{\mathbf{h}}_i^{(l+1)} = \mathbf{W}_{\text{concat}} \mathbf{m}_i^{\text{agg}}$
    Normalize and activate: $\mathbf{h}_i^{(l+1)} = \boldsymbol{\beta}_i \odot \mathbf{h}_i^{(l)} + (1 - \boldsymbol{\beta}_i) \odot \text{SiLU}(\text{BatchNorm}(\tilde{\mathbf{h}}_i^{(l+1)}))$
  **end for**
  **Physics Integration:**
    Compute form factors: $f_j(\mathcal{H}) = \text{MLP}_{\text{form}}(\mathbf{h}_j^{(L)})$ for $j = 1, \ldots, N$
    Calculate structure factors: $F_{hkl} = \sum_{j=1}^{N} f_j \exp(2\pi i \mathbf{h} \cdot \mathbf{r}_j)$ for $(h, k, l) \in \mathcal{H}, f_j \in f_j(\mathcal{H})$
    Construct diffraction map: $\mathbf{F}_{\text{concat}} = [\text{Re}(\mathbf{F}) \oplus \text{Im}(\mathbf{F})]$
  **Feature Fusion:**
    Pooling: $\mathbf{g} = \text{MeanPool}(\{\mathbf{h}_i^{(L)}\}_{i=1}^{N})$
    Process diffraction: $\mathbf{d} = \text{MLP}_{\text{diff}}(\mathbf{F}_{\text{concat}})$
    Fuse modalities: $\mathbf{z}_{\text{fused}} = \text{MLP}_{\text{fusion}}([\mathbf{g} \oplus \mathbf{d}])$
  **Final Prediction:** $\hat{y} = \text{MLP}_{\text{out}}(\mathbf{z}_{\text{fused}})$
  **return** $\hat{y}$

---

Table 8: Mean Absolute Error comparison on the JARVIS-DFT-3D-2021 dataset. We adopt the same data-splitting strategy as CrystalFormer (Taniai et al.), applying a random seed of *123* and dividing the dataset into training, validation, and test sets with an 8:1:1 ratio.

| **Method** | Form. Energy eV/atom | Bandgap (OPT) eV | Bulk Modulus log(GPa) | Shear Modulus log(GPa) | Total Energy eV/atom | Bandgap (MBJ) eV | $E_{\text{hull}}$ eV |
|---|---|---|---|---|---|---|---|
| CGCNN | 0.039 | 0.164 | 0.107 | 0.153 | 0.044 | 0.374 | 0.099 |
| SchNet | 0.048 | 0.219 | 0.109 | 0.162 | 0.067 | 0.459 | 0.198 |
| MEGNET | 0.045 | 0.175 | 0.120 | 0.201 | 0.048 | 0.409 | 0.089 |
| GATGNN | 0.042 | 0.201 | 0.107 | 0.155 | 0.051 | 0.450 | 0.079 |
| Matformer | 0.041 | 0.166 | 0.334 | 0.386 | 0.041 | 0.311 | 0.057 |
| Crystalformer | 0.032 | **0.125** | 0.077 | 0.141 | 0.034 | 0.305 | 0.060 |
| Crystalframer | 0.038 | 0.132 | 0.081 | 0.134 | 0.042 | 0.330 | 0.070 |
| eComFormer | 0.037 | 0.165 | 0.200 | 0.275 | 0.035 | 0.301 | 0.055 |
| EwaldMP | 0.059 | 0.209 | 0.120 | 0.169 | 0.084 | 0.451 | 0.120 |
| PotNet | 0.034 | 0.141 | 0.106 | 0.155 | 0.062 | 0.301 | 0.079 |
| ReGNet | 0.038 | 0.195 | 0.211 | 0.162 | 0.047 | 0.336 | 0.071 |
| **PRDNet** | **0.025** | 0.138 | **0.075** | **0.132** | **0.024** | **0.242** | **0.040** |

## D HYPERPARAMETERS

We provide a detailed hyperparameter settings used in the benchmark models for fair comparison.

- **CGCNN**: We use the released implementation from the official repo, with 300 training epochs, batch size of 64, an initial learning rate of 0.001 scheduled by OneCycleLR, and weight decay set to $1 \times 10^{-5}$. The graph encoder uses 256 atom features with 6 convolutional layers, followed by a fully connected network with hidden dimension 128. Mean squared error (MSE) loss is used for regression and negative log-likelihood (NLL) loss with log-softmax for classification.

Table 9: Mean Absolute Error comparison on the JARVIS-DFT-3D-2021 dataset. We adopt the baseline results reported by ReGNet (Nie et al., 2025) and Crystalframer (Ito et al., 2025).

| Method | Form. Energy eV/atom | Bandgap (OPT) eV | Total Energy eV/atom | Bandgap (MBJ) eV | $E_{hull}$ eV |
|---|---|---|---|---|---|
| CGCNN | 0.063 | 0.200 | 0.078 | 0.410 | 0.170 |
| SchNet | 0.045 | 0.190 | 0.047 | 0.430 | 0.140 |
| MEGNET | 0.047 | 0.145 | 0.058 | 0.340 | 0.084 |
| GATGNN | 0.047 | 0.170 | 0.056 | 0.510 | 0.120 |
| Matformer | 0.033 | 0.137 | 0.035 | 0.300 | 0.064 |
| Crystalformer | 0.031 | 0.128 | 0.032 | 0.274 | 0.046 |
| Crystalframer | 0.026 | **0.117** | 0.028 | 0.242 | 0.047 |
| eComFormer | 0.028 | 0.124 | 0.032 | 0.280 | 0.044 |
| PotNet | 0.029 | 0.127 | 0.032 | 0.270 | 0.055 |
| ReGNet | 0.027 | 0.126 | 0.027 | **0.240** | 0.043 |
| PRDNet | **0.025** | 0.138 | **0.024** | 0.242 | **0.040** |

Table 10: Mean Absolute Error comparison on the Materials Project (MEGNet) dataset. We adopt the baseline results reported by ReGNet (Nie et al., 2025).

| Method | Form. Energy eV/atom | Bandgap eV | Bulk Modulus log(GPa) | Shear Modulus log(GPa) |
|---|---|---|---|---|
| CGCNN | 0.031 | 0.292 | 0.047 | 0.077 |
| SchNet | 0.033 | 0.345 | 0.066 | 0.099 |
| MEGNET | 0.030 | 0.307 | 0.060 | 0.099 |
| GATGNN | 0.033 | 0.280 | 0.045 | 0.075 |
| Matformer | 0.021 | 0.211 | 0.043 | 0.073 |
| Crystalformer | 0.019 | 0.198 | 0.038 | 0.069 |
| Crystalframer | **0.017** | **0.185** | 0.034 | 0.068 |
| eComFormer | 0.018 | 0.202 | 0.042 | 0.073 |
| PotNet | 0.019 | 0.204 | 0.040 | 0.065 |
| ReGNet | **0.017** | 0.189 | 0.033 | 0.063 |
| PRDNet | 0.020 | 0.187 | **0.031** | **0.062** |

- **SchNet**: We implement SchNet in SchNetPack and the default training script, with 500 training epochs, batch size of 64. The graph was constructed with a neighbor cutoff of 8 Å and Gaussian radial basis functions with 80 bases. The graph encoder used hidden size of 128 and 3 interaction blocks. Optimization is performed with AdamW using a learning rate of $1 \times 10^{-3}$ and weight decay of $1 \times 10^{-5}$. Training is run with checkpointing on validation loss, without a learning rate scheduler. MSE loss is used for training regression tasks and the cross entropy loss for classification task.

- **MEGNet**: We adopted the MatGL implementation and the default training script. Crystal structures are converted into graphs with a neighbor cutoff of 4 Å and Gaussian radial basis expansion of 100 centers (width = 0.5). The embedding dimensions are 16 for nodes, 100 for edges, and 2 for global states. The network contains 3 MEGNet blocks with hidden sizes of (64, 32) in the input MLP, (64, 64, 32) in the convolutional MLP, and (32, 16) in the output MLP. The softplus2 is used as activation function. The model is trained with batch size 64 for up to 1000 epochs using Adam optimizer, mean squared error loss, and early stopping with patience of 500.

- **GATGNN**: We used the released implementation with graph attention layers. The network contains 4 GAT layers, each with 128 neurons and 4 attention heads. The default unpooling strategy is fixed-cluster. We train with a batch size of 64, learning rate of $1 \times 10^{-3}$, and AdamW optimizer with weight decay $1 \times 10^{-4}$. A MultiStepLR scheduler is applied with milestones at 150 and 250 epochs and decay factor 0.3. Training runs for up to 1000 epochs with SmoothL1 loss for regression tasks (CrossEntropyLoss for classification) and early stopping with patience of 150.

- **Matformer**: We use the released implementation from the official repo, with 500 training epochs, batch size of 512, an initial learning rate of 0.001 scheduled by OneCycleLR with 2000 warmup

steps, and weight decay set to $1 \times 10^{-5}$. The Matformer encoder uses 128 node features with 5 convolutional layers and 4 attention heads per layer, followed by a fully connected network with hidden dimension 128. The model employs 92-dimensional atom input features, 128-dimensional edge features, and uses k-nearest neighbor strategy with cutoff distance of 8.0 Å and maximum 12 neighbors. MSE loss is used for regression and NLL loss with log-softmax for classification.

- **Crystalformer**: We use the released implementation from the official repo, with 500 training epochs, batch size of 256, an initial learning rate of 0.0005 scheduled by inverse square root without warmup, and weight decay set to $1 \times 10^{-5}$. The CrystalFormer encoder uses 128 model dimensions with 4 transformer layers, 8 attention heads, and a feed-forward dimension of 512. The model employs adaptive cutoff with sigma value of -3.5, lattice range of 2, and uses both real and reciprocal space representations with Gaussian basis functions. Position encoding is applied with distance-based features (64 dimensions) and scale factors of 1.4 for real space and 2.2 for reciprocal space. L1 loss is used for regression tasks and cross-entropy loss for classification tasks, with average pooling for graph-level predictions.

- **Crystalframer**: We use the released implementation from the official repo, with 500 training epochs, batch size of 256, an initial learning rate of 0.0005 scheduled by inverse square root without warmup, and weight decay set to $1 \times 10^{-5}$. The CrystalFramer encoder uses 128 model dimensions with 4 transformer layers, 8 attention heads, and a feed-forward dimension of 512. The model employs adaptive cutoff with sigma value of -3.5, lattice range of 2, and uses real space representations with Gaussian basis functions. Position encoding is applied with distance-based features (64 dimensions) and angle-based features (64 dimensions), with scale factors of 1.4 for real space and 2.2 for reciprocal space. L1 loss is used for regression tasks and cross-entropy loss for classification tasks, with average pooling for graph-level predictions.

- **eComFormer**:We use the released implementation from the official repo, with 500 training epochs, batch size of 512, an initial learning rate of 0.001 scheduled by OneCycleLR with 2000 warmup steps, and weight decay set to 0. The eComformer encoder uses 256 node features with 3 convolutional layers and 1 attention head per layer, followed by a fully connected network with hidden dimension 256. The model employs 92-dimensional atom input features, 256-dimensional edge features, and uses k-nearest neighbor strategy with cutoff distance of 8.0 Å and maximum 12 neighbors. The architecture incorporates equivariant updates with second-order representations and uses radial basis functions with range from -4.0 to 0.0 for edge encoding. MSE loss is used for regression and cross-entropy loss for classification.

- **PotNet**: We use the released implementation from the official repo, with 100-300 training epochs, batch size of 64, an initial learning rate of $1 \times 10^{-3}$ optimized by AdamW, and no weight decay. A OneCycleLR scheduler with 2000 warmup steps is used to anneal the learning rate. Crystal structures are converted into graphs using 92-dimensional CGCNN atom features, a neighbor cutoff of 8 Å and at most 12 neighbors per atom. Local bond distances are expanded with a radial basis function (RBF) expansion into 256-dimensional edge features, while long-range periodic interactions are modeled by an infinite-summation branch over $R = 5$ shells using a combination of zeta and exponential potentials, which are further embedded into a 256-dimensional message-passing space. The PotNet encoder stacks 3 gated message-passing layers with 256 hidden channels, followed by global mean pooling and a fully connected network with hidden dimension 256 and ShiftedSoftplus activation. MSE loss is used for all regression targets.

- **EwaldMP**: We adapted the released implementation from the official repo for property prediction, with 100-300 training epochs, batch size of 64, an initial learning rate or 0.0005 (task dependent: 0.0001–0.001) scheduled by ReduceLROnPlateau with 0.8 fraction, 10 epochs patience, and 0.2 warmup steps. The AdamW optimizer was used with weight decay of 1e-6. Architecturally, we use the SchNet message passing network as backbone with 512 hidden channels, 256 filters, 4 interaction blocks, and 200 Gaussian radial basis functions, using a cutoff radius of 6.0 Å and at most 50 neighbors constructed under periodic boundary conditions. Long-range electrostatics are modeled by Ewald message passing module with reciprocal space on a $3 \times 3 \times 3$ voxel grid in $\{k\}$-space, applies Gaussian radial basis Fourier filters with a 16-dimensional linear bottleneck. The model uses additive pooling as readout, and MSE loss for regression and cross-entropy loss for classification tasks.

- **ReGNet/ReciNet**:We use the implementation with 100-300 training epochs, batch size of 64, an initial learning rate of 0.0008 (task-dependent: 0.0006-0.001) scheduled by OneCycleLR with

30% warmup steps, and weight decay set to 1e-5. The ReGNet architecture uses 304 hidden dimensions with 4 blocks, where each block integrates a short-range geometric GNN and a reciprocal space block for long-range interactions, followed by a fully connected decoder network with hidden dimensions [304, 304, 152]. The model employs 92-dimensional CGCNN atom input features projected to 304 dimensions, 304-dimensional RBF-expanded edge features with scaling factor -0.75/distance, and uses k-nearest neighbor strategy with maximum 16 neighbors. The architecture incorporates dual-space modeling with Fourier transforms in reciprocal space using fractional coordinates and reciprocal lattice vectors, and uses radial basis functions with 256 kernels ranging from -4.0 to 4.0 for edge encoding. MSE is used for regression and cross-entropy loss for classification.

## E    BACKGROUND OF PREDICTION TARGETS

- **Formation Energy**. The formation energy (Gillan, 1989) quantifies the thermodynamic stability of a material and is defined as the energy required to form a compound from its constituent elements in their reference states.

$$E_{\text{form}} = E_{\text{tot}} - \sum_i n_i \mu_i, \tag{83}$$

where $E_{\text{tot}}$ is the total energy of the compound, $n_i$ is the number of atoms of element $i$ in the formula unit, and $\mu_i$ is the chemical potential (reference energy) of element $i$. A negative formation energy indicates that the compound is thermodynamically favorable with respect to decomposition into its elements. The unit we used is eV/atom, obtained by dividing the total energy by the number of atoms.

- **Band Gap and Metal/Non-metal Classification** The band gap $E_{\text{g}}$(Zaanen et al., 1985) is a fundamental electronic property defined as the energy difference between the conduction band minimum (CBM) and the valence band maximum (VBM):

$$E_{\text{g}} = E_{\text{CBM}} - E_{\text{VBM}}. \tag{84}$$

The unit we used is eV. Materials with $E_{\text{g}} > 0$ are classified as semiconductors or insulators depending on the magnitude of the gap, whereas those with $E_{\text{g}} = 0$ are considered metallic. Hence, the band gap also serves as a basis for the **metal/non-metal classification** of materials.

- **Bulk Modulus**. The bulk modulus $B$ (Makishima & Mackenzie, 1975) represents resistance to uniform compression and is defined as:

$$B = -V \left( \frac{\partial P}{\partial V} \right), \tag{85}$$

where $V$ is the volume and $P$ is the pressure.

- **Shear Modulus**. The shear modulus $G$ (Makishima & Mackenzie, 1975) characterizes the material's response to shear stress and is calculated from the elastic stiffness tensor or averaged using the Voigt-Reuss-Hill approximation.

$$G = \frac{\tau}{\gamma}, \tag{86}$$

where $\tau$ is the shear stress and $\gamma$ is the shear strain. The unit we used is GPa, and we applied a logarithmic transformation to its values for comparison with the leaderboard results.

- **Young's Modulus**. The Young's modulus $E$ (Kalra et al., 2016) measures the stiffness under uniaxial stress. For isotropic materials, it is calculated as:

$$E = \frac{9BG}{3B + G}. \tag{87}$$

The unit we used is GPa, and we applied a logarithmic transformation to its values for comparison with the leaderboard results.

- **Exfoliation Energy**. The exfoliation energy (Jung et al., 2018) quantifies the energy cost to isolate a single atomic layer from a bulk layered material, which is critical in evaluating the feasibility of two-dimensional material synthesis. It is defined as:

$$E_{\text{exf}} = \frac{E_{\text{mono}} - E_{\text{bulk}}/N}{A},$$

(88)

where $E_{\text{mono}}$ is the energy of the isolated monolayer, $E_{\text{bulk}}$ is the total energy of the bulk material containing $N$ layers, and $A$ is the surface area of the monolayer. A lower exfoliation energy indicates easier mechanical or chemical exfoliation. The unit we used is meV/atom, obtained by dividing the exfoliation energy by the number of atoms.

