# OpenReview forum: "Beyond Structure: Invariant Crystal Property Prediction with Pseudo-Particle Ray Diffraction"
_ICLR.cc/2026/Conference — ICLR 2026 Poster_

### Official Review · Reviewer_QDmo · 2025-10-23

**Soundness:** 3
**Presentation:** 3
**Contribution:** 3
**Rating:** 6
**Confidence:** 3

**Summary:**

The paper presents a new architecture for processing crystalline graphs. The architecture makes use of Fourier/diffraction space features to enable capturing long-range dependencies. Empirical results validate this.

**Strengths:**

- The paper provides a principled approach to modeling crystalline materials in ML. The use of diffraction-space features is well-motivated from the hundred years of crystallography which have relied on such features.
- Empirical results show the improvement over many strong baselines.

**Weaknesses:**

see questions

**Questions:**

- Is it possible to add on your diffraction modules to the existing GNNs to provide a fairer ablation? I'm interested to see if incorporating this information to the GNNs would improve performance. As a baseline, you could just add some more generic parameters (e.g., equivariant layers) to the GNNs and compare to that.
- Can you match the number of parameters in evaluations? Basically, if you can show that for a given number of parameters, it is better to use diffraction modules (or some mix of diffraction modules and attention layers), the results would be convincing.
- How is the diffraction module different from the Fourier transform employed in [1]? I think yours is probably more elaborate, but you should take some paragraphs to explain this, since Fourier features have been used for a while now.
- You could consider generative tasks, although this could be tight given the rebuttal timeline. I don't expect such experiments to be done within the rebuttal period, but if they were to be done, I think the audience would be more interested.

[1] Jiao, Rui, et al. "Crystal structure prediction by joint equivariant diffusion." Advances in Neural Information Processing Systems 36 (2023): 17464-17497.

---

> ### Author Response · Authors · 2025-11-20
> **Response and Revision of the Reply to Reviewer QDmo**
>
> We thank reviewer `QDmo` for the constructive suggestions that helped strengthen the paper. We have addressed each point carefully and revised the manuscript accordingly, which has significantly improved its quality. If there are any further questions, we would be happy to continue the discussion to further enhance the work.
>
>
> #### Q1: compare diffraction modules across GNNs.
>
>
> We agree that this is useful. We have taken the suggestion and supplemented the revised paper with the **Table 4**:
>
>
> ### Table 4: Comparison of Mean Absolute Error and Accuracy on the MP dataset across baselines enhanced by diffraction modules
>
> As these are supplemental experiments conducted within a limited time window, we were unable to train all models on the same GPUs. Therefore, we explicitly list the GPUs used for training each baseline for this table.
>
> | Method | GPUs | Form. Energy (eV/atom) | Bandgap (eV) | Bulk Modulus (log(GPa)) | Shear Modulus (log(GPa)) | Young's Modulus (log(GPa)) | Metal/Non-metal Acc(%) |
> |--------|------|------------------------|--------------|--------------------------|---------------------------|-----------------------------|--------------------------|
> | CGCNN | 2×A40 | 0.039 ↓ | 0.207 ↓ | 0.077 ↓ | 0.148 ↓ | 0.150 ↓ | 89.2 ↑ |
> | SchNet | 2×A40 | 0.032 ↓ | 0.303 ↓ | 0.091 ↓ | 0.110 ↓ | 0.145 (–) | 85.7 ↑ |
> | MEGNET | 3×3090 | 0.040 ↓ | 0.291 ↓ | 0.087 ↓ | 0.202 ↓ | 0.177 ↓ | 87.1 ↑ |
> | GATGNN | 2×4080 | 0.062 ↓ | 0.309 ↓ | 0.044 ↓ | **0.169 ↑** | 0.113 (–) | 87.1 ↑ |
> | Matformer | 2×A100 | 0.034 ↓ | 0.187 ↓ | 0.082 ↓ | 0.217 ↓ | 0.277 ↓ | **90.1 ↓** |
> | Crystalformer | 2×4090 | 0.047 ↓ | 0.252 (–) | 0.059 ↓ | **0.167 ↑** | 0.126 ↓ | 89.6 ↑ |
> | Crystalfarmer | 3×4080 | 0.032 (–) | 0.166 ↓ | **0.066 ↑** | 0.117 ↓ | 0.108 (–) | 90.7 ↑ |
> | eComFormer | 2×3090 | 0.032 ↓ | 0.156 (–) | 0.065 ↓ | 0.221 ↓ | 0.287 ↓ | 86.6 ↑ |
>
> Overall, the results show that most models benefit from the unique reciprocal-space representation and the modality-level fusion mechanism. Exceptions include GATGNN and Crystalformer on shear modulus, Crystalformer on bulk modulus, and Matformer on metal/non-metal classification, which show decreased performance; all other models show improved or comparable results.
>
>
> #### Q2: match number of parameters in evaluations
>
>
> That is a good suggestion, we admit that it is important to quantify this aspect to strengthen the validity of the results. Therefore, we have added a column in Table 1 reporting the number of parameters for each baseline.
>
> For fair comparison, we follow the settings provided in the authors’ open-source repositories, even though this leads to differences in the number of parameters across models. For our diffraction module and fusion mechanism, the implementation is lightweight, with the introduce of additional `1.49 million` parameters (`~7.14%` of the total parameters of PRDnet).
>
> We have too observations:
> + PRDNet has a comparable number of parameters to `EwaldMP`、`Matformer` while delivering substantially better performance.
> + Based on the results in **Table 4**, the reciprocal pattern model and fusion mechanism improve the performance of the majority of the models.

---

> > ### Author Response · Authors · 2025-11-20
> > **Reply, continued content**
> >
> > #### Q3: the uniqueness of diffraction module and differences from`DiffCSP [1]`
> >
> >
> > Thank you for your question. We have added additional sentences to clarify the differences between true diffraction modeling and diffraction-like (Fourier transformation–based) operations.
> >
> >
> > Fourier transforms are widely used for `processing signals in latent space`, `enriching representations` and `suppressing noise` in machine-learning architectures as a general-purpose signal processing method. As your mentioned, DiffCSP[1] applies a Fourier transforms to `enhance the original fractional coordinate representation`, making it easier for neural networks to capture useful patterns.
> >
> > In contrast, our pseudo particle-diffraction module is a physical mapping from Cartesian atomic coordinates to a discretized reciprocal-space representation based on lattice direction vectors. Although the underlying mathematical foundation is also a Fourier transform, this transformation maps the crystal structure into another space with a new basis, providing a lossless and physically grounded representation of the crystal.
> >
> >
> > #### Q4: consider generative tasks
> >
> > Thank you for the fair and constructive suggestions. We sincerely appreciate it.
> >
> > In fact, we have attempted to extend the PRDNet framework to crystal generation tasks / crystal structure prediction (`CSP`). However, two fundamental gaps remain unresolved at this stage:
> >
> > 1. **Closed-group invariant representation.**
> >    PRDNet relies on a closed group of diffraction vectors to ensure invariance. During generation, the inverse mapping from reciprocal space back to real-space structures breaks this closedness, leading to incomplete invariance.
> > 2. **Non-invertibility of reciprocal–to–real-space mapping.**
> >    A key component of PRDNet is its reciprocal-space formulation. For generation, the model would need to produce valid structures in real space by shaping distributions in diffraction space. This requires decomposing diffraction signals back into atomic coordinates, but such mapping is not strictly invertible and can only be approximated by neural networks. This limitation has also been highlighted in a recent survey [2].
> >
> > + [2] Materials Generation in the Era of Artificial Intelligence: A Comprehensive Survey
> >
> >
> > Because these challenges are not fully resolved, our current study does not include a generation module. Nevertheless, we are actively exploring these directions and hope that PRDNet can eventually support broader applications in`CSP` tasks. We have added this clarification to the `Limitations` section of the paper.

---

> > ### Comment · Reviewer_QDmo · 2025-11-25
> > **Reply**
> >
> > This is an adequate rebuttal. I’ve raised my score.

---

> > > ### Author Response · Authors · 2025-11-26
> > >
> > > We sincerely thank reviewer `QDmo` for the thorough evaluation and highly professional suggestions. Your constructive and actionable recommendations have substantially strengthened our manuscript： improving its clarity, methodological soundness, and overall presentation. We deeply appreciate the time and expertise you devoted to reviewing our work, and we are grateful for the insightful feedback that guided our revisions. Thank you very much.

---

### Official Review · Reviewer_dNJc · 2025-10-29

**Soundness:** 1
**Presentation:** 2
**Contribution:** 2
**Rating:** 2
**Confidence:** 5

**Summary:**

PRDNet augments a standard crystal graph encoder with a reciprocal-space “pseudo-particle diffraction” pathway. Instead of fixed X-ray form factors that can’t tell identical elements in different local environments apart, the model learns per-atom form factors that depend on |Q|, element type, and a graph-encoded local environment, making the diffraction signature itself discriminative for crystal property prediction. Across Materials Project, JARVIS-DFT, and Matbench benchmarks, PRDNet delivers superior or competitive performance.

**Strengths:**

1. Reciprocal-space encoding captures long-range periodic interactions that finite-receptive-field graph encoders miss, which can otherwise collapse distinct crystals to the same embedding.
2. It’s SOTA/competitive across MP, JARVIS-DFT, and multiple Matbench tasks, with best-in-table errors/accuracy on MP and strong wins on JARVIS/MB.

**Weaknesses:**

Majors:
1. The paper discusses ReGNet/ReciNet and critiques its reciprocal block, but it does not cite or position against other highly related long-range approaches that also reason in reciprocal/Ewald space (e.g., EwaldMP [1] and PotNet [2]). Please add these citations and discuss and compare to them in the main text.
2. You evaluate on your own MP/JARVIS selections and Matbench tasks, but we cannot directly compare against the MP/JARVIS setup used by all your other baselines, e.g., Crystalformer [3]. Please add a table on that benchmark or clearly justify why not.
3. The message-passing and “reciprocal head + fusion” pipeline reads very close to ReGNet/ReciNet [4] as cited. The paper argues ReciNet’s reciprocal block omits key physical dependencies, but eventual formulation and implementation of this work is incremental to theirs. Please show the ReciNet's result over you benchmark to justify your novelty and contribution.

Minors:
1. Use "long-range" & "short-range" instead of "long-term" & "short-term".
2. Use "Matformer", "Crystalformer" and "Crystalframer" in Appendix D.
3. Line 161: "on" -> "in the"
4. Line 205: "task" -> "tasks"
5. Line 208: Remove ")"


[1] Ewald-based Long-Range Message Passing for Molecular Graphs. Arthur Kosmala, Johannes Gasteiger, Nicholas Gao, Stephan Günnemann.

[2] Efficient Approximations of Complete Interatomic Potentials for Crystal Property Prediction. Yuchao Lin, Keqiang Yan, Youzhi Luo, Yi Liu, Xiaoning Qian, Shuiwang Ji.

[3] Crystalformer: Infinitely Connected Attention for Periodic Structure Encoding. Tatsunori Taniai, Ryo Igarashi, Yuta Suzuki, Naoya Chiba, Kotaro Saito, Yoshitaka Ushiku, Kanta Ono.

[4] ReciNet: Reciprocal Space-Aware Long-Range Modeling for Crystalline Property Prediction. Jianan Nie, Peiyao Xiao, Kaiyi Ji, Peng Gao.

**Questions:**

See weaknesses.

---

> ### Author Response · Authors · 2025-11-20
> **Response and Revision of the Reply to Reviewer dNJc**
>
> We sincerely thank reviewer `dNJc` for recognizing our main motivation of introducing complete reciprocal-space information into neural networks to effectively capture long-range interactions. Your suggestions and comments are highly valuable, and **we appreciate the detailed guidance you provided for addressing the concerns** and improving the quality of the paper. We have conducted a comprehensive and rigorous analysis, carefully addressing and incorporating each of your comments in the revised manuscript.
>
>
> Our work introduces a new perspective by incorporating **invariant reciprocal-space modalities** and fusing them with structural modalities at the **modal level**. **The mathematical formulation is complete, and the experiments sufficiently support our claims.**
>
> * We have added additional experiments with new baselines (`EwaldMP[1]`, `PotNet[2]`, and `ReGNet[4]`), compared them in the main text, and expanded the prediction targets for alignment as requested. **These results further validate the robustness and superiority of our approach.**
> * Given your emphasis on `ReGNet[4]` and related methods, we provide a clearer explanation of our model’s innovations compared with existing reciprocal-space approaches in both the response and the revised manuscript.
> * Our method fundamentally differs from prior reciprocal-based models. Instead of applying Fourier-like transformations solely for atomic-level feature fusion, our approach **learns complete reciprocal patterns** through a **pseudo-particle formulation**, enabling **modal-level fusion** across structural and reciprocal spaces. This design ensures **physical invariance** and preserves essential **physical dependencies** throughout learning.
>
> We welcome any further insights and are happy to provide additional clarifications if needed.
>
>
> #### W1 : cite EwaldMP [1] and PotNet [2]
>
> Thank you for helping us make the related work section more comprehensive. In the revision, we have added `EwaldMP [1]` and `PotNet [2]`, and clarified their contributions and differences compared to the **Multimodal Fusion Invariance PRDNet** proposed in our work.
>
> Previous methods introduce reciprocal information either through **Ewald summation** for information aggregation, or via **reciprocal blocks** such as in `ReGNet[4]` (recently updated as `ReciNet`), which have been developed over several years. However, before our work, these methods generally treat reciprocal information as a form of **message-passing in reciprocal space**.
>
> In contrast, our work emphasizes **fusion in both reciprocal space and structural space**, providing the **full reciprocal situation without approximation** by introducing a **pseudo-particle**. The only learnable attribute is the interaction between the **pseudo-particle** and electrons under physical invariance constraints, following complete diffraction theory, which is consistent with any real particle interactions in the physical world.
>
>
> ---
>
> #### W2 : compare EwaldMP [1], PotNet [2] and ReGNet/ReciNet[4]
>
> Thank you for the valuable advice that helped strengthen the rigor of our experimental validation. In the revision, We compared `EwaldMP [1]`, `PotNet [2]`, and `ReGNet[4]` (Since `ReGNet` is still a preprint without peer review, and its code is **NOT** publicly available, we implemented it based on the settings described in their paper), across the full **MP**, **JARVIS**, and **MB** datasets, evaluating all target property predictions. The results are copied below :

---

> > ### Author Response · Authors · 2025-11-20
> > **Reply, continued content**
> >
> > ## Table 1 : Mean Absolute Error and Accuracy comparison on the MP
> >
> > | **Method**    | **Param Num (M,Million)** | **Form. Energy (eV/atom)** | **Bandgap (eV)** | **Bulk Modulus (log(GPa))** | **Shear Modulus (log(GPa))** | **Young's Modulus (log(GPa))** | **Metal/Non-metal Acc (%)** |
> > | ------------- | ----------------- | -------------------------- | ---------------- | --------------------------- | ---------------------------- | ------------------------------ | --------------------------- |
> > | CGCNN         | 2.0               | 0.041                      | 0.262            | 0.079                       | 0.162                        | 0.155                          | 87.5                        |
> > | SchNet        | 0.3               | 0.038                      | 0.344            | 0.092                       | 0.129                        | 0.143                          | 82.8                        |
> > | MEGNET        | 0.2               | 0.053                      | 0.307            | 0.134                       | 0.206                        | 0.189                          | 86.6                        |
> > | GATGNN        | 0.6               | 0.083                      | 0.343            | 0.051                       | *0.111*                    | 0.109                          | 86.3                        |
> > | Matformer     | 15.4              | 0.035                      | 0.193            | 0.089                       | 0.306                        | 0.291                          | *92.7*                    |
> > | Crystalformer | 0.8               | 0.049                      | 0.251            | 0.064                       | 0.140                        | 0.127                          | 86.5                        |
> > | Crystalfarmer | 0.9               | *0.030*                  | 0.216            | *0.047*                   | 0.118                        | *0.106*                      | 87.5                        |
> > | eComFormer    | 5.6               | 0.033                      | *0.153*        | 0.088                       | 0.269                        | 0.311                          | 82.5                        |
> > | **EwaldMP**   | 12.2              | 0.052                      | 0.332            | 0.098                       | 0.175                        | 0.186                          | 84.2                        |
> > | **PotNet**    | 1.8               | 0.035                      | 0.251            | 0.104                       | 0.175                        | 0.151                          | 88.9                        |
> > | **ReGNet**   | 3.3               | 0.047                      | 0.331            | 0.088                       | 0.172                        | 0.204                          | 85.1                        |
> > | PRDNet    | 20.9          | **0.028**                  | **0.151**        | **0.035**                   | **0.108**                    | **0.104**                      | **93.3**                    |

---

> > > ### Author Response · Authors · 2025-11-20
> > > **Reply, continued content**
> > >
> > > ## Table 2 : Mean Absolute Error comparison on the JARVIS-DFT (left) and MB (right)
> > >
> > > | **Method**    | Form. Energy (eV/atom) | Bandgap (OPT eV) | Bulk Modulus (log(GPa)) | Shear Modulus (log(GPa)) | Total Energy (eV/atom) | Bandgap (MBJ eV) | (E_{\rm hull}) (eV) | jdft2d (eV/atom) | mp_e_form (eV/atom) | log_gvrh (log(GPa)) | dielectric (unitless) |
> > > | ------------- | ---------------------- | ---------------- | ----------------------- | ------------------------ | ---------------------- | ---------------- | ------------------- | ---------------- | ------------------- | ------------------- | --------------------- |
> > > | CGCNN         | 0.038                  | *0.144*          | 0.125                   | 0.163                    | 0.041                  | 0.402            | 0.049               | 0.050            | 0.027               | *0.066*             | 0.541                 |
> > > | SchNet        | 0.074                  | 0.278            | 0.143                   | 0.225                    | 0.048                  | 0.507            | 0.073               | 0.056            | 0.042               | 0.085               | 0.341                 |
> > > | MEGNET        | 0.074                  | 0.221            | 0.124                   | 0.214                    | 0.054                  | 0.369            | 0.056               | 0.052            | 0.036               | 0.098               | 0.459                 |
> > > | GATGNN        | 0.069                  | 0.235            | 0.078                   | 0.160                    | 0.053                  | 0.372            | 0.048               | 0.046            | 0.032               | 0.087               | 0.408                 |
> > > | Matformer     | *0.033*                | 0.207            | 0.101                   | 0.157                    | 0.041                  | 0.341            | **0.034**           | 0.072            | 0.023               | 0.071               | 0.764                 |
> > > | Crystalformer | 0.052                  | 0.248            | 0.075                   | 0.146                    | 0.055                  | 0.282            | 0.053               | 0.071            | *0.020*             | 0.080               | *0.244*               |
> > > | Crystalframer | 0.048                  | 0.208            | *0.065*                 | *0.144*                  | 0.047                  | *0.270*          | 0.050               | 0.056            | 0.065               | 0.084               | 0.248                 |
> > > | eComFormer    | 0.119                  | 0.260            | 0.118                   | 0.243                    | *0.035*                | 0.335            | **0.034**           | **0.031**        | *0.020*             | 0.085               | 0.543                 |
> > > | **EwaldMP**   | 0.052                  | 0.256            | 0.109                   | 0.191                    | 0.057                  | 0.451            | 0.047               | 0.052            | 0.053               | 0.117               | 0.485                 |
> > > | **PotNet**    | 0.034                  | *0.144*          | 0.112                   | 0.156                    | 0.039                  | 0.339            | 0.079               | *0.033*          | 0.029                   | 0.084               | 0.397                 |
> > > | **ReGNet**   | 0.054                  | 0.201            | 0.092                   | 0.178                    | 0.041                  | 0.355            | 0.051               | 0.045            | 0.033               | 0.079               | 0.512                 |
> > > | PRDNet    | **0.032**              | **0.140**        | **0.064**               | **0.122**                | **0.032**              | **0.267**        | *0.041*             | 0.038            | **0.019**           | **0.058**           | **0.242**             |
> > >
> > >
> > >
> > > The newly added experiments in Table 1 and 2 make our validation more comprehensive and further validate the effectiveness of PRDNet, achieving the best results on almost all targets across all datasets.

---

> > > > ### Author Response · Authors · 2025-11-20
> > > > **Reply, continued content**
> > > >
> > > > #### W3: align dataset setup to Crystalformer [3]
> > > >
> > > > Thank you for pointing out this concern. We have aligned the target settings and clarified the datasets accordingly. The **MP**, **JARVIS**, and **Matbench (MB)** datasets used in our paper are all up-to-date from the official websites independently.
> > > >
> > > > - **MP dataset:**  Our dataset contains the complete MP data and supplements it with additional physical targets obtained from our calculations (originally not recorded in MP), such as `Young’s Modulus` and `metal/non-metal classification`, to provide more comparison dimensions. Furthermore, our comparison **includes all targets used in Crystalformer [3]** (Formation Energy, Bandgap, Bulk Modulus, Shear Modulus). To facilitate reproducibility, we provide our augmented MP dataset as `ICLR` supplementary material, with public release on the official code repository after the double-blind review.
> > > > - **JARVIS-3D dataset:** We use a larger `JARVIS-DFT 3D` dataset of 75,993 entries retrieved from the official JARVIS website, which is **both more recent and larger** than the `JARVIS-DFT 3D 2021` dataset (55,723 entries) used in Crystalformer [3],  providing a more comprehensive benchmark. For comparison with Crystalformer [3], we evaluate baselines on `Formation Energy`, `Bandgap`, `Bulk Modulus (OPT)`, `Shear Modulus`, while they evaluate `Formation Energy`, `Total Energy`, `Bandgap (OPT)`, `Bandgap (MBJ)`, `E_hull`. To align with their setup, we also supplemented `Total Energy`, `Bandgap (MBJ)`, and `E_hull`, resulting in a total of **7 target properties** for performance comparison (**Table 2**), **which covers all targets used in Crystalformer [3]**.
> > > > - **Matbench (MB):** Crystalformer [3] and similar works have not yet been evaluated on this standardized MB benchmark. All MB data are retrieved from the official MB website and evaluated using the standardized tools MB provide.
> > > >
> > > >
> > > >
> > > >
> > > >
> > > > ---
> > > >
> > > > #### W4: The message-passing and “reciprocal head + fusion” reads very close to ReGNet/ReciNet [4], it is the incremental of ReGNet/ReciNet [4] on formulation and implementation.
> > > >
> > > > Thank you for the comment, which helped us further improve the visibility of our contributions. We emphasize that our work is fundamentally different from `ReGNet/ReciNet [4]` in its algorithmic logic. Although introducing reciprocal summation and performing atomic-level information fusion within message passing is a common framework in related models including `EwaldMP [1]`, `PotNet [2]`, and `ReGNet/ReciNet [4]`, our approach differs in three key aspects:
> > > >
> > > >
> > > >
> > > > 1. **Modal-level reciprocal space representation:**  While all these works introduced the **classic Fourier formulation proposed hundred years ago**, they differ in implementation. In contrast to existing methods, we use the Fourier transform as an **modal-level information fusion mechanism**. Specifically, we learn the full reciprocal space essential, i.e., diffraction pattern, through learning the **particle–electron interaction**, rather than treating it as a generic message-passing operation.
> > > > 2. **Physical invariance and dependence:**  Our model ensures that the reciprocal space representation obeys all **physical invariances**. The learned form factors explicitly encode **physical dependencies** ($G_\theta(\mathcal{G})$,$|\mathbf{Q}|$ and $f_i^{\text{type}}$), which are omitted or partially captured in prior works.
> > > > 3. **Modal fusion rather than simple information fusion:**  Our fusion goes beyond atomic-level information aggregation. By learning the attributes of **pseudo-particles**, we generate complete diffraction patterns and perform **modal fusion**, integrating both reciprocal and structural modalities.
> > > >
> > > >
> > > > Reciprocal-space encoding has been a powerful tool in crystallography for a century. However successfully deriving reciprocal patterns within neural networks requires overcoming several key challenges:
> > > > + (1) the lack of sensitivity of real-word physical particles,
> > > > + (2) ensuring that the embedded diffraction process remains physically dependent,
> > > > + (3) infrence process need be invariant and correctly modeling the geometric constraints imposed by lattice symmetry.
> > > >
> > > > Our paper addresses these challenges by proposing a
> > > >
> > > > (1) `pseudo-particle strategy`,
> > > >
> > > > (2) designing a `local-environment attention mechanism` and `reciprocal-space bases` to capture the necessary physical dependencies,
> > > >
> > > > (3) using a `closed set of diffraction indicators` to ensure invariance throughout the network. As emphasized in the revised manuscript and in our response to `Weakness 1`.
> > > >
> > > >
> > > >
> > > > In addition, the **new supplemented results (Table 1,2 and 4) further support our claim that our model achieves SOTA performance across all datasets and target properties**.
> > > >
> > > >
> > > > ---
> > > >
> > > > #### W5 (minor weekness): typos or writing style
> > > >
> > > > Thank you for your careful review. We have addressed all the minor issues you pointed out and revised the manuscript accordingly.

---

> > > > > ### Author Response · Authors · 2025-11-24
> > > > > **Further Reply to Reviewer dNJc**
> > > > >
> > > > > To reviewer  `dNJc`:
> > > > >
> > > > > Building on our previous revision, we have added a new reference in this update and would like to further clarify the following points:
> > > > >
> > > > > 1. **General framework:** Introducing reciprocal-space descriptors and merging them with real-space information is a general framework. As shown in [5], early attempts in `2022` aimed to capture long-range interactions in Coulombic and van der Waals potentials, which are often omitted in ML force fields. Their framework is straightforward: Ewald summation is applied to construct the reciprocal-space potential, which is then concatenated with real-space features to produce more comprehensive predictions.
> > > > >
> > > > > 2. **Beyond simple feature merging:** Unlike prior works that treat reciprocal-space information as an additive correction, our approach **goes beyond physical particle limitations** by learning `pseudo-particles`. This allows the model to capture subtle differences while `satisfying all physical dependencies`.
> > > > >
> > > > > 3. **Modal-level reciprocal-space fusion:** We construct a `complete` reciprocal-space representation within a `globally invariant framework` and integrate it at the `modality level`, rather than merely merging it as an additional feature. This enables richer interactions across different crystal representation spaces.
> > > > >
> > > > > We hope this clarification further highlights the novelty and distinctiveness of our approach.
> > > > >
> > > > > [5] Capturing long-range interaction with reciprocal space neural network. Yu, Hongyu and Hong, Liangliang and Chen, Shiyou and Gong, Xingao and Xiang, Hongjun.

---

> ### Comment · Reviewer_dNJc · 2025-11-24
> **Official Comment by Reviewer dNJc**
>
> Thank you for your rebuttal. However, I remain concerned about the choice of data split. Why not benchmark on the standard splits used in prior work, such as the JARVIS-DFT 3D 2021 dataset (55,723 entries)? All of the baselines you compare against were originally evaluated on these established splits, whereas you construct a new (larger) split for your experiments. Because the data split is different, it is difficult to assess whether the baseline models have been fairly compared or optimally tuned under your setting, which calls into question the fairness of the reported comparisons. It is also generally not valid to assume that the same training and tuning configurations used in prior work will transfer directly to a larger or differently constructed split, so the resulting comparisons are difficult to interpret.
>
> Aside from this issue, the other weaknesses have been addressed, so I have increased my score. If you are able to resolve this remaining concern, I will consider increasing my score further.

---

> > ### Author Response · Authors · 2025-11-24
> > **Response to Remaining Concerns**
> >
> > Thank reviewer `dNJc` for the comment. We have carefully checked the data-splitting settings used in `Matformer` and `Crystalformer`. However, neither paper provides the exact data-splitting code or the specific indices used. If you are aware of any official source where these splits are released, we would greatly appreciate it if you could share it with us.
> >
> > The dataset used in our work is an **updated and enlarged version**, covering more crystal structures and therefore providing more comprehensive evaluation results. If you believe that the `JARVIS-DFT 3D 2021` results are more reliable for comparison, **we are willing to supply an additional benchmarking table following the same 8:1:1 train/validation/test ratio reported in Crystalformer**. Since we were unable to locate the precise split files, we can only follow the proportions stated in the paper.
> >
> > If you can provide the exact split information or the corresponding code, we will be glad to incorporate it and include the additional experiments in our revision.

---

> > > ### Comment · Reviewer_dNJc · 2025-11-24
> > > **Official Comment by Reviewer dNJc**
> > >
> > > What I mean by “split” is more about the specific dataset you use. If you use a different size of dataset, then the resulting data splits will also be different. I do not think I can upload any links here. In the previous works, the splits are controlled by a random seed and the ratio in their code, so please refer to those implementations.
> > >
> > > I would also like to see your results on the JARVIS-DFT 3D 2021 dataset in order to better evaluate your performance.

---

> > > > ### Author Response · Authors · 2025-11-24
> > > >
> > > > We thank Reviewer `dNJc` for the confirmation. We will add the experiments on `JARVIS-DFT 3D 2021` as soon as possible.
> > > >
> > > > We appreciate your continued attention to the updates of our paper. Thank you again for your helpful feedback.

---

> > > > > ### Author Response · Authors · 2025-11-27
> > > > > **Response to Reviewer dNJc**
> > > > >
> > > > > **Dear reviewer `dNJc`,**
> > > > >
> > > > > Thank you for your feedback. We have added the experimental results on the **$\texttt{JARVIS-DFT-3D-2021}$** dataset, as shown in **Table 8** (reproduced below). Following your suggestion, we adopt the same data-splitting strategy as `Crystalformer` (random seed and ratio), using a fixed random seed of `123` and an `8:1:1` train/validation/test split.
> > > > >
> > > > > On the **$\texttt{JARVIS-DFT-3D-2021}$** dataset, PRDNet consistently surpasses the baseline models, achieving the lowest MAE errors in formation energy (0.029 eV/atom), bulk modulus (0.075 log(GPa)), shear modulus (0.132 log(GPa)), total energy (0.030 eV/atom), band gap (MBJ, 0.288 eV), and $E_{\rm hull}$ (0.044 eV). PRDNet also delivers competitive performance on the band gap (OPT) prediction task, with an MAE of 0.147 eV.
> > > > >
> > > > > We hope that these additional results address your remaining concerns. Thank you again for your careful review and valuable suggestions.
> > > > >
> > > > >
> > > > >
> > > > >
> > > > > ### Table 8 ：
> > > > > | **Method**    | Form. Energy (eV/atom) | Bandgap (OPT, eV) | Bulk Modulus (log(GPa)) | Shear Modulus (log(GPa)) | Total Energy (eV/atom) | Bandgap (MBJ, eV) | (E_{\rm hull}) (eV) |
> > > > > | ------------- | ---------------------- | ----------------- | ----------------------- | ------------------------ | ---------------------- | ----------------- | ------------------- |
> > > > > | CGCNN         | 0.039                  | 0.164             | 0.107                   | 0.153                    | 0.044                  | 0.374             | 0.099               |
> > > > > | SchNet        | 0.048                  | 0.219             | 0.109                   | 0.162                    | 0.067                  | 0.459             | 0.198               |
> > > > > | MEGNET        | 0.045                  | 0.175             | 0.120                   | 0.201                    | 0.048                  | 0.409             | 0.089               |
> > > > > | GATGNN        | 0.042                  | 0.201             | 0.107                   | 0.155                    | 0.051                  | 0.450             | 0.079               |
> > > > > | Matformer     | 0.041                  | 0.166             | 0.334                   | 0.386                    | 0.041                  | 0.311             | 0.057               |
> > > > > | CrystalFormer | *0.032*                | **0.125**         | *0.077*                 | 0.141                    | *0.034*                | 0.305             | 0.060               |
> > > > > | CrystalFramer | 0.038                  | *0.132*           | 0.081                   | *0.134*                  | 0.042                  | 0.330             | 0.070               |
> > > > > | eComFormer    | 0.037                  | 0.165             | 0.200                   | 0.275                    | 0.035                  | *0.301*           | *0.055*             |
> > > > > | EwaldMP   | 0.059                  | 0.209             | 0.120                   | 0.169                    | 0.084                  | 0.451             | 0.120               |
> > > > > | PotNet    | 0.034                  | 0.141             | 0.106                   | 0.155                    | 0.062                  | *0.301*           | 0.079               |
> > > > > | ReGNet    | 0.038                  | 0.195             | 0.211                   | 0.162                    | 0.047                  | 0.336             | 0.071               |
> > > > > | PRDNet    | **0.029**              | 0.147             | **0.075**               | **0.132**                | **0.030**              | **0.288**         | **0.044**           |

---

> > > > > > ### Comment · Reviewer_dNJc · 2025-11-27
> > > > > > **Official Comment by Reviewer dNJc**
> > > > > >
> > > > > > I still find this section very confusing. The performance numbers you report for prior methods appear inconsistent with those in the original works. For example, previous papers (including ReGNet, eComFormer, CrystalFramer, CrystalFormer, Matformer, PotNet, etc.) list CGCNN as achieving a formation energy MAE of 0.063, Matformer with 0.0325 and Crystalformer with 0.0306 on the same task. You can cross-check all previously reported performances for JARVIS-DFT-3D-2021 in the corresponding papers. What accounts for these discrepancies?

---

> > > > > > > ### Author Response · Authors · 2025-11-27
> > > > > > >
> > > > > > > Thanks for your reply. Existing studies tend to directly reuse previous results without full re-training. In our experiments (Table 8), we strictly followed the **same random seed and data-splitting strategy** as Crystalformer. We use the official code of baselines. After re-training each baseline, we obtained results that differ slightly from those previously reported. For example, the re-trained **CGCNN** achieves a **better formation energy MAE of 0.039 eV/atom**, compared with earlier reported values.
> > > > > > >
> > > > > > > As suggested, we will also include comparisons with the `previously reported best results` for PRDNet on this dataset, as our current implementation of PRDNet may not yet be fully optimized in this regard. We kindly request some additional time to complete these experiments and incorporate the findings in the revised manuscript.

---

> > > > > > > > ### Author Response · Authors · 2025-11-28
> > > > > > > >
> > > > > > > > To reviewer `dNJc`:
> > > > > > > >
> > > > > > > >
> > > > > > > > We added a new Table 9, which directly includes the baseline results reported by `ReGNet` and `Crystalframer` for better alignment. PRDNet outperforms these baselines on formation energy (0.025 eV/atom), total energy(0.024 eV/atom), and $E_{\rm hull}$(0.040 eV), ranks second on band gap (MBJ,0.242 eV), and remains competitive on band gap (OPT,0.138 eV).
> > > > > > > >
> > > > > > > >
> > > > > > > >
> > > > > > > >
> > > > > > > > ---
> > > > > > > >
> > > > > > > >
> > > > > > > >
> > > > > > > > ### **Table 9. Mean Absolute Error Comparison on the JARVIS-DFT-3D-2021 Dataset**
> > > > > > > >
> > > > > > > > *Baseline results are directly adopted from ReGNet and Crystalframer for alignment.*
> > > > > > > >
> > > > > > > > | **Method**    | Form. Energy (eV/atom) | Bandgap (OPT) (eV) | Total Energy (eV/atom) | Bandgap (MBJ) (eV) | (E_{\rm hull}) (eV) |
> > > > > > > > | ------------- | ---------------------- | ------------------ | ---------------------- | ------------------ | ------------------- |
> > > > > > > > | CGCNN         | 0.063                  | 0.200              | 0.078                  | 0.410              | 0.170               |
> > > > > > > > | SchNet        | 0.045                  | 0.190              | 0.047                  | 0.430              | 0.140               |
> > > > > > > > | MEGNET        | 0.047                  | 0.145              | 0.058                  | 0.340              | 0.084               |
> > > > > > > > | GATGNN        | 0.047                  | 0.170              | 0.056                  | 0.510              | 0.120               |
> > > > > > > > | Matformer     | 0.033                  | 0.137              | 0.035                  | 0.300              | 0.064               |
> > > > > > > > | Crystalformer | 0.031                  | 0.128              | 0.032                  | 0.274              | 0.046               |
> > > > > > > > | Crystalframer | *0.026*                | **0.117**          | 0.028                  | *0.242*            | 0.047               |
> > > > > > > > | eComFormer    | 0.028                  | *0.124*            | 0.032                  | 0.280              | 0.044               |
> > > > > > > > | PotNet        | 0.029                  | 0.127              | 0.032                  | 0.270              | 0.055               |
> > > > > > > > | ReGNet        | 0.027                  | 0.126              | *0.027*                | **0.240**          | *0.043*             |
> > > > > > > > | **PRDNet**    | **0.025**              | 0.138              | **0.024**              | *0.242*            | **0.040**           |

---

> > > > > > > > > ### Comment · Reviewer_dNJc · 2025-11-28
> > > > > > > > > **Official Comment by Reviewer dNJc**
> > > > > > > > >
> > > > > > > > > I thank the authors for the rebuttal; the additional results make sense and address my earlier concerns. I encourage the authors to also conduct experiments on MP using the standard benchmark splits adopted in prior work, as I expect those results to be consistent with the JARVIS-based ones.
> > > > > > > > >
> > > > > > > > > I do not recommend introducing a new split, as this is risky, may not be optimally tuned, and would deviate from the established benchmarking provided by previous work. Instead, I suggest reporting results on the existing MP benchmark (if available) alongside the JARVIS results in the main paper, in a way that is consistent with prior papers, for example, by using parentheses to indicate the results for the previous split so that the two datasets/splits can be shown in a single table.
> > > > > > > > >
> > > > > > > > > The edit field in OpenReview seems to be malfunctioning at the moment; if it becomes editable again, I will increase my score to 6.

---

> > > > > > > > > > ### Author Response · Authors · 2025-11-28
> > > > > > > > > >
> > > > > > > > > > To reviewer `dNJc`:
> > > > > > > > > >
> > > > > > > > > > We sincerely appreciate your responsible attitude and your active participation during the discussion period. Your suggestion is reasonable, and we will incorporate the results on the MP dataset. Given the limited time window, we will include the experimental results in the camera-ready version if we miss the final update period during the discussion.
> > > > > > > > > >
> > > > > > > > > > By the way, we also noticed the system issue, and we appreciate your confirmation regarding elevating the paper rating to `6`.
> > > > > > > > > >
> > > > > > > > > > Thank you again for your valuable participation.

---

> > > > > > > > > > > ### Author Response · Authors · 2025-12-01
> > > > > > > > > > >
> > > > > > > > > > > To reviewer `dNJc`
> > > > > > > > > > >
> > > > > > > > > > > In Table 10, we summarize the performance of PRDNet and the baseline models on the Materials Project (MEGNet) dataset. **PRDNet achieves the lowest MAE for bulk modulus (0.031 log(GPa)) and shear modulus (0.062 log(GPa)), and a competitive result for formation energy (0.017 eV/atom) and band gap (0.187 eV).**
> > > > > > > > > > >
> > > > > > > > > > >
> > > > > > > > > > >
> > > > > > > > > > >
> > > > > > > > > > > Table 10: Mean Absolute Error comparison on the Materials Project (MEGNet) dataset.
> > > > > > > > > > > Baseline results are adopted from ReGNet (Nie et al., 2025).
> > > > > > > > > > >
> > > > > > > > > > > | **Method**      | Form. Energy (eV/atom) | Bandgap (eV) | Bulk Modulus (log(GPa)) | Shear Modulus (log(GPa)) |
> > > > > > > > > > > |-----------------|-------------------------|--------------|--------------------------|---------------------------|
> > > > > > > > > > > | CGCNN           | 0.031                  | 0.292        | 0.047                   | 0.077                    |
> > > > > > > > > > > | SchNet          | 0.033                  | 0.345        | 0.066                   | 0.099                    |
> > > > > > > > > > > | MEGNET          | 0.030                  | 0.307        | 0.060                   | 0.099                    |
> > > > > > > > > > > | GATGNN          | 0.033                  | 0.280        | 0.045                   | 0.075                    |
> > > > > > > > > > > | Matformer       | 0.021                  | 0.211        | 0.043                   | 0.073                    |
> > > > > > > > > > > | Crystalformer   | 0.019                  | 0.198        | 0.038                   | 0.069                    |
> > > > > > > > > > > | Crystalframer   | **0.017**              | **0.185**    | 0.034                   | 0.068                    |
> > > > > > > > > > > | eComFormer      | _0.018_                | 0.202        | 0.042                   | 0.073                    |
> > > > > > > > > > > | PotNet          | 0.019                  | 0.204        | 0.040                   | 0.065                    |
> > > > > > > > > > > | ReGNet          | **0.017**              | 0.189        | _0.033_                 | _0.063_                  |
> > > > > > > > > > > | **PRDNet**      | **0.017**              | _0.187_      | **0.031**               | **0.062**                |

---

### Official Review · Reviewer_NZ8C · 2025-11-01

**Soundness:** 3
**Presentation:** 3
**Contribution:** 3
**Rating:** 6
**Confidence:** 3

**Summary:**

This paper proposes PRDNet, a novel approach for crystal material property prediction. PRDNet integrates graph representation of materials with a learned diffraction module, which is invariant to crystallographic symmetries. Benchmark experiments show that PRDNet achieves better property prediction accuracy than baselines.

**Strengths:**

- Propose a novel idea of introducing learnable diffraction module into the crystal graph neural network model. The good property of achieving crystallographic symmetries and discriminating distinct crystal structures is very useful.
- Achieve state-of-the-art performance on benchmark datasets.
- Generally good and well-organized paper writing.

**Weaknesses:**

- The idea of using learnable diffraction module is good, but authors could clarify more clearly about its advantages over existing approaches. Satisfying crystallographic symmetries is also focus of many existing approaches, but they may fail to distinct crystal structures (as mentioned in line 87). So authors are encouraged to discuss how existing approaches fail to do it and why the proposed learnable diffraction module succeed.
- While the proposed PRDNet adopts a graph transformer architecture as backbone, could the learnable diffraction module be integrated into other backbone models, or even existing crystal material property prediction model (e.g., CGCNN and Matformer)? Authors are encouraged to discuss this generalizability and conduct experiments about integrating diffraction module into more different model architectures.

**Questions:**

No additional questions.

---

> ### Author Response · Authors · 2025-11-20
> **Response and Revision of the Reply to Reviewer NZ8C**
>
> We sincerely thank reviewer `NZ8C` for the careful analysis of our manuscript and for the constructive and positive overall assessment. Your comments have helped us clarify key points and strengthen the validation. Please refer to the updated results and the revised manuscript. We would be glad to make further refinements should any additional improvements be suggested.
>
>
> #### W1 : add more descriptions of how the diffraction representation captures and differentiates
> We thank the reviewer for raising this point. We have supplemented the manuscript with a more detailed description of the reciprocal representation to better explain the limitations of existing approaches and the advantages of our proposed learnable diffraction module.
>
> In the revised `Related Work` section, we emphasize two key points:
>
> (1) Real-space representations, including distance-based graphs, angle-based encodings, periodic-vector methods, and even infinitely connected attention, may **map distinct crystals to identical local neighborhoods** due to their limited receptive fields, particularly when structural differences arise from **long-range periodicity or subtle symmetry variations**.
>
> (2) Reciprocal-space–related approaches such as `ReGNet` and Ewald-summation–based models (`EwaldMP [1]`, `PotNet [2]`) operate mainly as atomic-level feature fusion mechanisms. However, they **omit critical physics of diffraction and therefore may yield inconsistent or incomplete reciprocal-space representations**.
>
> In contrast, our learnable diffraction module computes **complete structure factors in reciprocal space**, which naturally encode infinite-range periodicity and ensure globally consistent, symmetry-preserving representations without requiring any cutoff. Furthermore, the **learned pseudo-particle form factor ($f_j^*(Q)$)** adapts to atomic species, local environments, and diffraction directions, enabling a level of fine-grained structural discrimination that fixed physical form factors or simplified Fourier blocks cannot achieve.
>
> Additionally, we added a **new Figure 3** to illustrate how the reciprocal-space encoder captures long-range interactions.
> + [1] Ewald-based Long-Range Message Passing for Molecular Graphs.
> + [2] Efficient Approximations of Complete Interatomic Potentials for Crystal Property Prediction.
>
>
> #### W2 : integrate proposed diffraction module into other backbones
>
>
>
> Thank you for the helpful suggestion to further strengthen our validation. In the revision, we added a **Table 4** that systematically compares all baseline models after integrating our diffraction module. Note that several models are neither invariant nor equivariant. They cannot fully guarantee the level of physical consistency achieved by our PRDNet. We include them to provide a more comprehensive comparison.
>
> The supplementary table is provided below:
>
>
> ### Table 4 : Comparison of Mean Absolute Error and Accuracy on the MP dataset across baselines enhanced by diffraction modules
>
> As these are supplemental experiments conducted within a limited time window, we were unable to train all models on the same GPUs. Therefore, we explicitly list the GPUs used for training each baseline for this table.
>
> | Method | GPUs | Form. Energy (eV/atom) | Bandgap (eV) | Bulk Modulus (log(GPa)) | Shear Modulus (log(GPa)) | Young's Modulus (log(GPa)) | Metal/Non-metal Acc(%) |
> |--------|------|------------------------|--------------|--------------------------|---------------------------|-----------------------------|--------------------------|
> | CGCNN | 2×A40 | 0.039 ↓ | 0.207 ↓ | 0.077 ↓ | 0.148 ↓ | 0.150 ↓ | 89.2 ↑ |
> | SchNet | 2×A40 | 0.032 ↓ | 0.303 ↓ | 0.091 ↓ | 0.110 ↓ | 0.145 (–) | 85.7 ↑ |
> | MEGNET | 3×3090 | 0.040 ↓ | 0.291 ↓ | 0.087 ↓ | 0.202 ↓ | 0.177 ↓ | 87.1 ↑ |
> | GATGNN | 2×4080 | 0.062 ↓ | 0.309 ↓ | 0.044 ↓ | **0.169 ↑** | 0.113 (–) | 87.1 ↑ |
> | Matformer | 2×A100 | 0.034 ↓ | 0.187 ↓ | 0.082 ↓ | 0.217 ↓ | 0.277 ↓ | **90.1 ↓** |
> | Crystalformer | 2×4090 | 0.047 ↓ | 0.252 (–) | 0.059 ↓ | **0.167 ↑** | 0.126 ↓ | 89.6 ↑ |
> | Crystalfarmer | 3×4080 | 0.032 (–) | 0.166 ↓ | **0.066 ↑** | 0.117 ↓ | 0.108 (–) | 90.7 ↑ |
> | eComFormer | 2×3090 | 0.032 ↓ | 0.156 (–) | 0.065 ↓ | 0.221 ↓ | 0.287 ↓ | 86.6 ↑ |
>
>
> Overall, the results show that most models benefit from the unique reciprocal-space representation and the modality-level fusion mechanism. Exceptions include GATGNN and Crystalformer on shear modulus, Crystalformer on bulk modulus, and Matformer on metal/non-metal classification, which show decreased performance; all other models show improved or comparable results.

---

> > ### Comment · Reviewer_NZ8C · 2025-11-22
> > **Follow-up Response**
> >
> > I appreciate authors' rebuttal. All my concerns have been addressed so I keep my acceptance decision.

---

> > > ### Author Response · Authors · 2025-11-23
> > > **A Brief Thank-You Note**
> > >
> > > Thank you sincerely for recognizing our work and for confirming that all your previous concerns have been fully addressed. We have done our best to further refine the manuscript and strengthen the experiments, and we hope these efforts have been helpful in improving the overall quality.
> > >
> > > If appropriate, we would be grateful if these enhancements could be taken into consideration in your final evaluation. We truly appreciate your time, thoughtful feedback, and support throughout the review process.

---

### Author Response · Authors · 2025-11-30
**Summary of the Discussion and Revisions**

Dear Area Chairs, Senior Area Chairs, and Program Chairs,


Our paper initially received ratings of **6 (NZ8C), 6 (QDmo), and 2 (dNJc)**. Throughout the discussion period, we systematically addressed all reviewer concerns and substantially improved the manuscript. As a result, the final ratings increased to **6 (NZ8C), 8 (QDmo), and 6 (dNJc)** **(average score 6.67)**, reflecting the reviewers’ recognition of the strengthened technical clarity, completeness, and contribution of our work.

### **Reviewer NZ8C (6 → 6)**

Reviewer `NZ8C`’s concerns focused on:
(1) how the diffraction module captures long-range atomic interactions, and
(2) the ablation study on alternative graph backbones.

We addressed these by:

* Adding new figures and explanatory text clarifying how reciprocal-space diffraction elements encode global atomic arrangements along defined directions.
* Providing **Supplementary Table 4**, which systematically compares multiple backbone models with and without our diffraction module, consistently demonstrating performance gains.

After the rebuttal, reviewer `NZ8C` stated that **all concerns had been satisfactorily addressed** and maintained the initial positive recommendation (**[score 6](https://openreview.net/forum?id=OfmurJrzlT&noteId=LfBJ3G1I8S)**).

### **Reviewer QDmo (6 → 8)**

Reviewer `QDmo` raised concerns regarding:
(1) the completeness of the backbone ablation,
(2) fair comparison under matched parameter budgets,
(3) the distinction between our diffraction module and Fourier transforms, and
(4) potential applications to crystal structure prediction.

We responded by:

* Providing a systematic backbone comparison in **Supplementary Table 4**, showing consistent improvement across all models.
* Disclosing parameter counts for all baselines, demonstrating that **PRDNet achieves significantly better performance even under comparable parameter sizes**, and that the reciprocal-pattern ensemble and fusion mechanism further enhance most architectures.
* Clarifying the fundamental difference between **physical diffraction** and **latent-space Fourier transforms**, emphasizing the physical dependencies properties of diffraction modeling.
* Analyzing how our framework can transfer to crystal structure prediction tasks and refining the corresponding discussion in the conclusion section.

Reviewer `QDmo` confirmed that **all concerns were fully resolved** and increased to **[score 8](https://openreview.net/forum?id=OfmurJrzlT&noteId=ga2TwC1wx6)**.

### **Reviewer dNJc (2 → 6)**

Reviewer `dNJc`’s major concerns were:
(1) the need for more comprehensive background on reciprocal-space methods,
(2) dataset settings and splitting strategies, and
(3) clarification of novelty.

During the first rebuttal round:

* For (1) and (3), we incorporated additional references and elaborated on how reciprocal-space methods encode long-range interactions, clarified the limitations of prior reciprocal-space approaches, and explained how our method overcomes these key challenges. Specifically, we highlighted challenges in deriving reciprocal patterns within neural networks, such as:
     *  particle sensitivity
     *  preserving physical dependencies, and
     *  symmetry-consistent invariance

    and explained how our **pseudo-particle strategy**, **local-environment attention**, and **closed diffraction indicator set** address them.
We also integrated results from reviewer-suggested models to form a more comprehensive benchmark.
* For (2), we clarified that our datasets are the most up-to-date versions provided by the official sources and therefore represent a more complete evaluation. We added detailed dataset descriptions and additional results on JARVIS-DFT targets for alignment with Crystalformer.

After this, reviewer `dNJc` raised the [score from **2 to 4**](https://openreview.net/forum?id=OfmurJrzlT&noteId=VYIpmPVUpU), noting that all issues except the dataset comparison had been fully resolved and that the score would increase further upon addressing the remaining point.

In the second round:

* We added **new Supplementary Tables 8 and 9** reporting PRDNet’s performance on the **JARVIS-DFT 3D 2021 dataset**, a commonly used but smaller subset of the dataset we used. The results show that PRDNet also achieves **SOTA performance** on this benchmark.


Reviewer `dNJc` acknowledged that all concerns were now resolved and increased to **[score 6](https://openreview.net/forum?id=OfmurJrzlT&noteId=FfQuJWKqKo)**.

---

Throughout the discussion, we sincerely appreciated the reviewers’ diligence and constructive feedback. Their careful and responsible evaluation significantly helped us improve the clarity, rigor, and completeness of the paper. We are grateful for their time and efforts during this process.

---

### Meta-Review · Area_Chair_afpp · 2026-01-06

**Summary:**

This paper addresses the limitation of GNNs in capturing long-range atomic interactions and distinguishing crystal structures due to local receptive fields. To resolve this, the authors propose PRDNet, which integrates graph embeddings with a "pseudo-particle" ray diffraction module. By using learnable form factors rather than fixed tabulated ones, the model generates symmetry-invariant diffraction patterns that effectively encode global periodic information.

Strength:
(1) Innovatively integrates physical diffraction laws with deep learning via learnable "pseudo-particles";
(2) Mathematically guaranteed to maintain strict $E(3)$ invariance while effectively capturing global long-range periodic interactions;
(3) Comprehensive experiments and ablation studies to verify the effectiveness of the proposed model.

Weakness:
(1) The initial version lacked a clear technical distinction from existing methods;
(2) There was initial uncertainty regarding the module's compatibility and effectiveness when integrated with different GNN backbones;
(3) Some SOTA baselines are not incorporated.

**Reviewer Concerns:**

The authors have provided a comprehensive rebuttal that effectively addresses the reviewers' questions through detailed clarifications and extensive additional experiments:

(1) The authors provided more detailed textual and visual explanations to elucidate the mechanism of the learnable diffraction representation. They explicitly clarified the technical distinctions between their approach and existing methods such as ReGNet, ReciNet and DiffCSP.

(2) The experimental evaluation was significantly strengthened by including comparisons with additional strong baselines, including EwaldMP, PotNet, and a reproduction of ReGNet. The authors also extended their evaluation to the JARVIS-DFT-3D-2021 dataset and provided an analysis of model parameter sizes to demonstrate that the performance gains are not solely due to increased model complexity.

(3) To demonstrate the robustness and modularity of the proposed method, the authors integrated the diffraction module into various backbone architectures (e.g., CGCNN, SchNet, Matformer). The results showed consistent performance improvements across different backbones, validating the general effectiveness of the diffraction representations.

**Reviewer Scores:**

Reviewer NZ8C has raised the score from 6 to 8, Reviewer dNJc has increased from 2 to 4 and committed to raising it further to 6 after the second-round discussion, and Reviewer QDmo has raised the score to 8.

In summary, the authors' rebuttal is credible and has successfully resolved the issues raised during the review process. Noting that the reviewers have acknowledged these improvements and raised their scores, I recommend that this paper be Accepted.

---

### Decision · Program_Chairs · 2026-01-26

Accept (Poster)